# Topological Alignment of Shared Vision-Language Embedding Space

**Junwon You**
Department of Mathematics
POSTECH, Republic of Korea
jwyou627@postech.ac.kr

**Dasol Kang**
Dololo Research Engineer
english4118@gmail.com

**Jae-Hun Jung**
Department of Mathematics
POSTECH, Republic of Korea
jung153@postech.ac.kr

## Abstract

Contrastive Vision-Language Models (VLMs) have demonstrated strong zero-shot capabilities. However, their cross-modal alignment remains biased toward English due to limited multilingual multimodal data. Recent multilingual extensions have alleviated this gap but enforce instance-level alignment while neglecting the global geometry of the shared embedding space. We address this problem by introducing **ToMCLIP** (**To**pological Alignment for **M**ultilingual **CLIP**), a topology-aware framework aligning embedding spaces with topology-preserving constraints. The proposed method applies persistent homology to define a topological alignment loss and approximates persistence diagram with theoretical error bounds using graph sparsification strategy. This work validates the proposed approach, showing enhanced structural coherence of multilingual representations, higher zero-shot accuracy on the CIFAR-100, and stronger multilingual retrieval performance on the xFlickr&CO. Beyond VLMs, the proposed approach provides a general method for incorporating topological alignment into representation learning.

## 1 INTRODUCTION

Contrastive Vision-Language Models (VLMs), such as CLIP (Radford et al., 2021) and ALIGN (Jia et al., 2021) have demonstrated strong zero-shot transfer capabilities by learning a shared embedding space for images and texts (Bordes et al., 2024). These models align paired samples through contrastive learning, enabling diverse downstream tasks without task-specific supervision. Although autoregressive multimodal large language models such as LLaVA (Liu et al., 2024c), Qwen-VL (Bai et al., 2023), and Gemini (Team et al., 2023) have recently achieved vision-language understanding via generative training, contrastive VLMs remain effective for retrieval tasks and computational efficiency.

Despite recent multilingual extensions (Carlsson et al., 2022; Chen et al., 2023; Yang et al., 2024), representation spaces remain structurally misaligned. Most approaches enforce instance-level alignment via distillation or continual learning, but they fail to preserve the global geometry in the shared embedding space. This structural misalignment causes unstable cross-lingual retrieval and inconsistent semantic clustering.

As illustrated in Figure 1, the English and Korean text embeddings produced by the CLIP encoder are not aligned. Even the multilingual CLIP (MCLIP; Carlsson et al., 2022) fails to achieve cross-lingual

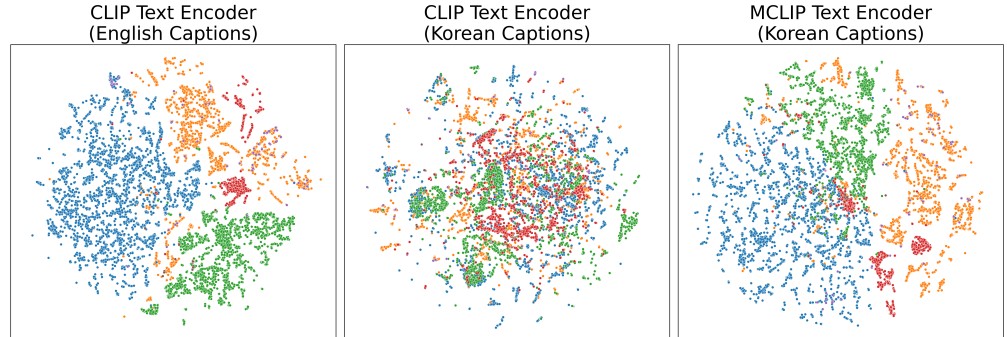

Figure 1: Visualization of text embeddings (English and Korean) in the latent space using t-SNE (Maaten and Hinton, 2008), from CLIP and multilingual CLIP (MCLIP; Carlsson et al., 2022) text encoders. The Fashion Product Images dataset (Aggarwal, 2019) was used, where the `productDisplayName` field serves as the input caption to the text encoders. Colors indicate the corresponding `masterCategory` of each product.

alignment, with multiple semantic categories remaining intermixed in the center of the embedding space. To address this limitation, we propose **ToMCLIP**: **To**pological Alignment for **M**ultilingual **CLIP**, a topology-aware training framework that enforces structural consistency across languages using topological data analysis. This approach is motivated by the hypothesis that performance gaps between English and other languages stem from differences in the topological structure of their latent representations.

The contributions of this work are as follows:

- We introduce a topology-aware training framework for multilingual contrastive VLMs. It formalizes the structural misalignment across languages and addresses it with a topological alignment loss that enforces structural alignment in the shared embedding space.

- We develop a scalable approximation for persistence diagrams. The approach constructs sparse graphs using MST-based sparsification and provides theoretical error bounds of approximation.

- We validate the proposed method using case studies on multilingual vision-language tasks. The experiments reveal improved cross-lingual structural coherence, higher zero-shot accuracy on the CIFAR-100, and stronger multilingual retrieval performance on the xFlickr&CO.

Appendix A reviews related work on contrastive VLMs, autoregressive multimodal large language models and topological analysis of the embedding space.

## 2 TOPOLOGICAL ALIGNMENT

Figure 2 presents an overview of our proposed alignment framework. Appendix B presents the preliminaries of the persistent homology, including persistence diagrams and the (sliced) Wasserstein distance.

We integrated topological alignment loss with MCLIP. The MCLIP proposes a teacher-student framework that applies machine-translated captions for training. A set of English captions $X$ is translated into a target language to form $X^*$. The CLIP text encoder $E_T$ (teacher) encodes the original captions $X$, and the MCLIP text encoder $E_S$ (student) encodes the translated captions $X^*$. Then $E_S$ is trained to align with the teacher by minimizing the mean squared error (MSE) between the output embeddings:

$$L_{\text{pw}} = \text{MSE}(E_T(X), E_S(X^*)). \tag{1}$$

This approach focuses on point-wise alignment, overlooking the structural consistency of the embedding space across languages.

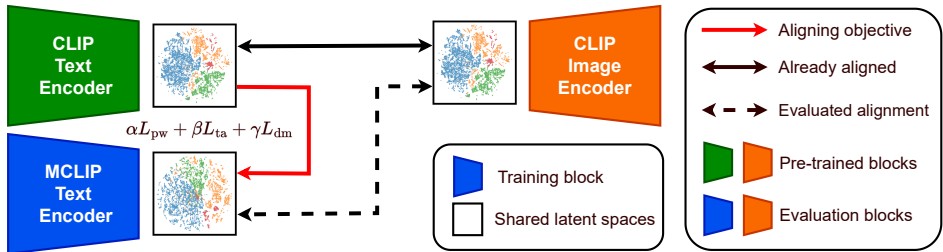

Figure 2: Overview of the proposed alignment framework between CLIP ($E_T$) and multilingual CLIP (MCLIP; $E_S$) text encoders. $E_S$ is trained to align with the frozen $E_T$ using a combination of loss functions; $L_{\text{pw}}$ enforces point-wise alignment; $L_{\text{ta}}$ and $L_{\text{dm}}$ promote geometric alignment by preserving topological structures. The evaluation is conducted by pairing $E_S$ with the pretrained CLIP image encoder, enabling cross-lingual retrieval in the shared embedding space.

## 2.1 Topological Alignment Loss

We introduce a novel topological alignment loss $L_{\text{ta}}$ that enforces the global structural alignment. For a batch of data comprising $N$ image-text pairs $\{(I_i, T_i)\}_{i=1}^N$, the text representations $\{E_T(T_i)\}_{i=1}^N$ form a geometric structure in the embedding space (Figure 1). The MCLIP loss $L_{\text{pw}}$ considers each representation $E_T(T_i)$ independently, ignoring the geometric relationships between the samples.

To address this problem, we compute the persistence diagram $D_T$ from the point cloud $\{E_T(T_i)\}_{i=1}^N$, which summarizes the topological features of the embedding distribution (e.g., connected components and cycles). Similarly, we compute $D_S$ from the point cloud $\{E_S(T_i^*)\}_{i=1}^N$, where $T_i^*$ denotes the translated caption of $T_i$, capturing the structure of the MCLIP. To align these spaces, we define the topology alignment loss:

$$L_{\text{ta}} = \text{SW}_p^{(K)}(D_T, D_S), \tag{2}$$

where $\text{SW}_p$ denotes the sliced $p$-Wasserstein distance (SWD, Bonneel et al., 2015) and $K$ represents the number of projection directions. The SWD provides a fast, differentiable, and GPU-friendly approximation of the Wasserstein distance, making it suitable as a training loss. Minimizing the discrepancy between $D_T$ and $D_S$ enforces both embedding spaces to preserve comparable global topological structures, complementing pointwise matching with structural alignment.

Furthermore, we define a distance matrix loss $L_{\text{dm}}$ to promote local geometric alignment between the latent spaces. Given a point cloud $X = \{x_i\}_{i=1}^N$, the pairwise distance matrix is computed as follows: $(M_X)_{i,j} = \|x_i - x_j\|$ for $1 \leq i, j \leq N$ where $\|\cdot\|$ denotes the Euclidean ($L_2$) norm. The distance matrix loss is defined as follows:

$$L_{\text{dm}} = \text{MSE}(M_T, M_S) \tag{3}$$

where $M_T$ and $M_S$ denote the distance matrices computed from the point clouds $T = \{E_T(T_i)\}_{i=1}^N$ and $S = \{E_S(T_i^*)\}_{i=1}^N$, respectively. The total training objective is defined as the weighted sum of three loss components: $L_{\text{total}} = \alpha L_{\text{pw}} + \beta L_{\text{ta}} + \gamma L_{\text{dm}}$, where $\alpha$, $\beta$, and $\gamma$ are hyperparameters that control the relative contributions of each loss term.

**Stability-Based Justification of the Loss Design.** Let $X, Y \subset \mathbb{R}^n$ be finite point clouds. The $k$-dimensional persistence diagrams are denoted by $D_X^{(k)}$ and $D_Y^{(k)}$, respectively. By the stability theorem, for any $p \geq 1$, $C_k \geq 1$ exists such that

$$W_p\big(D_X^{(k)}, D_Y^{(k)}\big) \leq C_k W_p^c(X, Y), \tag{4}$$

where $W_p$ is the $p$-Wasserstein distance between diagrams and $W_p^c$ is the $p$-Wasserstein distance between point clouds (Skraba and Turner, 2020). Thus, if $W_p\big(D_X^{(k)}, D_Y^{(k)}\big) \geq \tau$, then $W_p^c(X, Y) \geq \tau/C_k$. Therefore, minimizing the distance between persistence diagrams ($\mathcal{L}_{\text{ta}}$) reduces the certified lower bound on the point cloud discrepancy. Moreover, because $D^{(0)}$ summarizes the connectivity in the embedding space, minimizing $\mathcal{L}_{\text{ta}}$ between $D^{(0)}$s reduces the cross-lingual semantic cluster misalignment, encouraging semantically equivalent texts to belong to the same cluster.

However, $\mathcal{L}_{\text{ta}}$ and $\mathcal{L}_{\text{dm}}$ are invariant to Euclidean isometries. If $Y = RX + t$ with $R \in O(n)$ (i.e., $R^\top R = I$ and $\det R \in \{\pm 1\}$) and $t \in \mathbb{R}^n$, then $\mathcal{L}_{\text{ta}} = \mathcal{L}_{\text{dm}} = 0$ and $W_p^c(X, Y)$ can be arbitrarily

large. Hence, these terms alone do not reduce $W_p^c$ or prevent rigid-motion drift. Therefore, $\mathcal{L}_{\text{pw}}$ is needed to fix the coordinate frame, while $\mathcal{L}_{\text{ta}}$ aligns the global topology and $\mathcal{L}_{\text{dm}}$ matches the pairwise geometry.

## 2.2 Approximating Persistence Diagrams

This work employs two strategies to approximate the persistence diagram of the Vietoris-Rips (Rips) complex with reduced computational overhead:

- We restrict the computation to 0-dimensional ($H_0$) features and the birth times of 1-dimensional ($H_1$) features, which can be extracted from the minimal spanning tree (MST) (Kruskal, 1956) with a union-find (Tarjan, 1979). This eliminates the need to construct the full Rips complex. Prior work has confirmed that $H_0$ features are sufficient to capture the topological structure of latent representations (Moor et al., 2020; Kim et al., 2024).
- To reduce the computational cost of MST further, we build a sparse graph from pairwise distances between embeddings, limiting the number of candidate edges.

This approximation reduces memory and time, enabling persistence diagrams in large-scale training. For a point cloud with $N$ points, computing the Rips complex has an exponential complexity of up to $\mathcal{O}(N^{k+1})$ for $k$-dimensional simplices. Persistent homology via boundary-matrix reduction has a worst case time of $\mathcal{O}(m^3)$ and a memory of $\mathcal{O}(m^2)$ (Otter et al., 2017), where $m$ denotes the total number of simplices in the filtration. Consequently, computing $H_0$ has a cost of $m = \mathcal{O}(N^2)$ up to $\mathcal{O}(N^6)$, whereas computing $H_1$ costs $m = \mathcal{O}(N^3)$ up to $\mathcal{O}(N^9)$. However, $H_0$ and the birth time of $H_1$ features can be computed via the MST, which has a computational complexity of $\mathcal{O}(E \log V)$, where $V$ denotes the number of vertices and $E$ represents the number of edges (Cormen et al., 2022). Notably, for $H_0$, only $N - 1$ edges are necessary to determine the death time, corresponding to the edges of the MST, out of a total $\binom{N}{2}$ edges in the fully connected graph. Therefore, constructing the MST over a complete graph is computationally inefficient. To mitigate this problem, we construct a sparse graph $G_\epsilon = (V, E_\epsilon)$ from a point cloud $X = \{x_1, \cdots, x_N\} \subset (\mathbb{R}^n, d)$, where $V = \{x_i\}_{i=1}^N$ and $E_\epsilon = \{(x_i, x_j) \mid d(x_i, x_j) \leq \epsilon\}$, with $d$ denoting a metric (e.g., Euclidean distance). This sparsification reduces the number of edges while retaining a sufficient topological structure to approximate the persistence diagram.

We calculate the upper bound on the approximation error of the proposed method. We construct a weighted complete graph $G = (V, E, \omega)$ from a point cloud $X$, where $V = X$, $E = \{(x_i, x_j) \mid x_i, x_j \in X, i \neq j\}$, and the weight function $\omega : E \to \mathbb{R}_{\geq 0}$ is defined as

$$\omega((x_i, x_j)) = \frac{d(x_i, x_j)}{M}, \tag{5}$$

where $M = \max\limits_{(x_i, x_j) \in E} d(x_i, x_j)$. By construction, $0 \leq \omega(e) \leq 1$ for all $e \in E$.

**Theorem 1.** *Let $0 \leq \epsilon \leq 1$ and $G_\epsilon = (V, E, \omega_\epsilon)$,*

$$\omega_\epsilon(e) = \begin{cases} \omega(e), & \text{if } \omega(e) \leq \epsilon, \\ 1, & \text{if } \omega(e) > \epsilon. \end{cases} \tag{6}$$

*Let $m(\epsilon) := \#\{(0, d) \in D_0^{\text{Rips}}(G) \mid \epsilon < d < \infty\}$, i.e., the number of finite 0-dimensional persistence points of $G$ whose death times exceed $\epsilon$. Then,*

$$W_p\big(D_0^{\text{Rips}}(G), D_0^{\text{Rips}}(G_\epsilon)\big) \leq m(\epsilon)^{1/p}(1 - \epsilon) \tag{7}$$

*and $0 \leq m(\epsilon) \leq N - 1$ where $W_p$ denotes the $p$-Wasserstein distance.*

Appendix C presents the proof of this theorem. Let $c(\epsilon)$ denote the number of connected components in $\text{VR}_\epsilon(G)$ which is equal to $m(\epsilon) + 1$. Therefore,

$$W_p\big(D_0^{\text{Rips}}(G), D_0^{\text{Rips}}(G_\epsilon)\big) \leq (c(\epsilon) - 1)^{1/p}(1 - \epsilon). \tag{8}$$

As $\epsilon$ increases, more edges are retained, sparsity decreases, and the number of connected components $c(\epsilon)$ monotonically decreases. In particular, a critical value $\epsilon_*$ exists such that $c(\epsilon) = 1$ for all

Table 1: Top-10 accuracy (%) of zero-shot classification on CIFAR-100 across 13 languages (Full vs. Low).

| Setting | Model | Languages (13) | | | | | | | | | | | | | *Avg* |
|---|---|---|---|---|---|---|---|---|---|---|---|---|---|---|---|
| | | En | Fr | Es | De | It | Ru | Pl | Tr | Da | Ja | Zh | Ko | Vi | |
| **Full** | CLIP | 91.06 | 66.18 | 63.69 | 64.05 | 49.33 | 11.95 | 22.03 | 24.73 | 32.42 | 32.80 | 21.56 | 12.38 | 15.32 | 39.04 |
| | MCLIP | 91.97 | 85.66 | 87.10 | 85.74 | 88.23 | 87.98 | **85.38** | 87.65 | 87.83 | 53.60 | 89.50 | 87.20 | 86.26 | 84.93 |
| | ToMCLIP($L_{dm}$) | **91.99** | 84.77 | 84.63 | **89.63** | 86.17 | 87.78 | 84.86 | 87.35 | 86.88 | 56.27 | 88.11 | 87.94 | 86.98 | 84.87 |
| | ToMCLIP($L_{ta}$) | 91.48 | 85.41 | 84.23 | 87.85 | 88.49 | **89.43** | 84.35 | **88.76** | 87.98 | 58.57 | **89.75** | 88.76 | **89.41** | 85.73 |
| | ToMCLIP | 91.40 | **87.59** | **87.37** | 89.30 | **89.11** | 87.66 | 83.59 | 88.59 | 87.79 | 57.95 | 88.68 | 88.36 | 88.17 | **85.81** |
| **Low** | CLIP | **91.06** | 66.18 | 63.69 | 64.05 | 49.33 | 11.95 | 22.03 | 24.73 | 32.42 | 32.80 | 21.56 | 12.38 | 15.32 | 39.04 |
| | MCLIP | 79.72 | 67.60 | 62.20 | 71.41 | 59.68 | 69.80 | 64.55 | 58.71 | 73.31 | 60.68 | 78.27 | 65.43 | 71.38 | 67.90 |
| | ToMCLIP($L_{dm}$) | 79.46 | 67.99 | 62.51 | 70.81 | 60.75 | 69.30 | 64.02 | 57.21 | 72.64 | 59.20 | 77.43 | 67.42 | 70.07 | 67.60 |
| | ToMCLIP($L_{ta}$) | 80.00 | 67.37 | 62.66 | 70.09 | 60.88 | 70.31 | 65.22 | 59.50 | 72.68 | 60.94 | 77.36 | 67.01 | **73.37** | 68.26 |
| | ToMCLIP | 80.75 | **68.56** | **63.85** | **71.49** | **62.91** | **71.23** | **65.50** | **60.80** | **73.75** | **62.39** | **78.82** | **67.96** | 72.44 | **69.26** |

$\epsilon \geq \epsilon_*$, (i.e., $\text{VR}_\epsilon(G)$ becomes connected). From an algorithmic perspective, the critical trade-off lies in selecting $\epsilon$ so that $\text{VR}_\epsilon(G)$ remains sparse while maintaining a small number of connected components. The experiments confirm that moderate values of $\epsilon$ already achieve near connectivity with a low edge density, making the sparsification highly effective in practice (Section F.1).

## 3 RESULTS

We evaluate ToMCLIP under two training conditions: (1) using the full available dataset and (2) using only 1% of the data for the low-resource setting. This setup is designed to mimic realistic situations where only a few of multilingual annotated data are available for training. Appendix D provides details on dataset preparation, training and evaluation. The ToMCLIP($L_{dm}$), ToMCLIP($L_{ta}$), and ToMCLIP denote models trained with the proposed total loss $L_{total}$ using the coefficients $(\alpha, \beta, \gamma) = (1, 0.01, 0)$, $(1, 0, 0.01)$, and $(1, 0.01, 0.01)$, respectively.

### 3.1 Evaluation on CIFAR-100

We evaluate the zero-shot classification on CIFAR-100 to assess the alignment between the image and multilingual text embeddings. At inference, we use class-name prompts translated into 13 languages (e.g., *"a photo of a {class}"*). Appendix E presents the complete prompt list. Table 1 reports the Top-10 accuracy (%) per language (the Top-1 and Top-5 are provided in Tables 6 and 7 in Appendix F.2). In the **Full** setting, ToMCLIP surpasses MCLIP in all but one language (Polish, "Pl"), yielding a higher average Top-10 accuracy overall ($+0.88$). In the **Low** setting, ToMCLIP outperforms MCLIP across all 13 languages ($+1.36$ on average). Note that En in the **Low** does not indicate catastrophic forgetting: CLIP's text encoder is not used when evaluating (To)MCLIP. Although MCLIP provides multilingual support, its cross-modal alignment remains suboptimal, whereas preserving the topological structure enables ToMCLIP to deliver more robust and consistent multilingual representations. Table 5 (Appendix F.2) summarizes the average Top-$k$ ($k \in \{1, 5, 10\}$) accuracy. The ToMCLIP performs the best for all $k$ and both regimes. Among the ablations, ToMCLIP($L_{dm}$) matches MCLIP, whereas ToMCLIP($L_{ta}$) consistently improves upon MCLIP. Using both losses together, ToMCLIP yields the strongest results. Adding $L_{dm}$ on the baseline $L_{pw}$ alone does not yield additional cross-modal alignment, whereas $L_{ta}$ alone induces extra alignment and improves accuracy. Nevertheless, $L_{dm}$ is beneficial in conjunction with $L_{ta}$, suggesting a complementary role that reinforces the alignment signal provided by $L_{ta}$. Appendices F.4 - F.8 presents the ablation studies on batch size, loss coefficients, homology dimension, graph sparsification threshold, and the number of SWD projections $K$, respectively.

## 4 CONCLUSION

This work introduces ToMCLIP, a topology-aware alignment framework for multilingual contrastive VLMs, augmenting instance-level matching with topology-preserving objectives. The ToMCLIP improves the zero-shot CIFAR-100 performance, and stronger multilingual retrieval performance on the xFlickr&CO. Furthermore, ToMCLIP enhances the structural coherence of the shared embedding space. Beyond multilingual alignment, the topological alignment loss provides a general objective for aligning embedding spaces, encompassing cross-modal alignment, knowledge distillation, and dimensionality reduction.

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

# Contents

# A  RELATED WORKS

## A.1  Contrastive Vision-Language Models

Contrastive vision-language models (VLMs) learn joint representations of images and text by maximizing the similarity between matched pairs while minimizing it for unmatched pairs. CLIP (Contrastive Language-Image Pre-training) (Radford et al., 2021) pioneered this approach by training dual encoders on 400 million image-text pairs collected from the internet. The model employs a symmetric cross-entropy loss over the similarity matrix of image and text embeddings within each batch, enabling zero-shot transfer to downstream tasks without task-specific fine-tuning.

ALIGN (Jia et al., 2021) scaled this approach further by leveraging a noisy dataset of over one billion image-text pairs, demonstrating that the noise in web-scraped data can be overcome with sufficient scale. Unlike CLIP, which uses curated data, ALIGN shows that raw alt-text data can be effective when combined with a simple dual-encoder architecture and contrastive learning objective.

Several subsequent works have improved upon these foundations. FLIP (Li et al., 2023b) introduced a masking strategy during training to reduce computational costs while maintaining performance. DeCLIP (Li et al., 2022) enhanced data efficiency through self-supervised learning and nearest-neighbor supervision. FILIP (Yao et al., 2022) improved fine-grained alignment by introducing token-wise maximum similarity between image patches and text tokens.

The key advantages of contrastive models include: (1) computational efficiency during inference, as image and text encoders can be cached and indexed separately; (2) flexibility in swapping encoders for different modalities or languages; and (3) strong performance on retrieval tasks. These properties make contrastive models particularly suitable for multilingual extensions, as the text encoder can be replaced or fine-tuned for different languages while keeping the image encoder fixed.

Despite their success, contrastive models face challenges in maintaining consistency across languages when extended to multilingual settings, particularly in preserving the geometric structure of the shared embedding space. We address this limitation through topological alignment.

## A.2  Multilingual Extensions of Contrastive VLMs.

Various multilingual extensions of contrastive VLMs have been developed, using knowledge distillation, continual learning, or multilingual pretraining to align images and texts across languages. For example, MCLIP (Carlsson et al., 2022) trains a single multilingual text encoder using text-only, machine-translation-based distillation to match the original CLIP English text space. In contrast, mCLIP (Chen et al., 2023) retains the dual-encoder design of CLIP but aligns a multilingual text encoder to CLIP via Triangle Cross-modal Knowledge Distillation (TriKD). The multilingual text encoder is initialized using contrastive pretraining. Continual language learning approaches (Yang et al., 2024) add languages incrementally to mitigate catastrophic forgetting.

## A.3  Autoregressive Multimodal Large Language Models

While our work focuses on contrastive VLMs, we briefly review recent autoregressive multimodal Large Language Models (LLMs) to contextualize our approach within the broader landscape of vision-language understanding. Unlike contrastive models that learn aligned embedding spaces, autoregressive multimodal LLMs generate text conditioned on visual inputs through next-token prediction.

Flamingo (Alayrac et al., 2022) pioneered the frozen LLM approach by introducing cross-attention layers between a pretrained vision encoder and language model, enabling few-shot learning on vision-language tasks. BLIP-2 (Li et al., 2023a) proposed Q-Former, a lightweight module that bridges frozen image encoders and LLMs through a set of learnable query tokens, significantly reducing training costs while achieving strong performance.

LLaVA (Liu et al., 2024c) demonstrated that visual instruction tuning (training on instruction-following data in the visual domain) can produce capable multimodal assistants. The model uses a simple projection layer to connect CLIP visual features with an LLM, showing that architectural simplicity combined with high-quality instruction data can be highly effective. Subsequent versions

like LLaVA-1.5 (Liu et al., 2024a) and LLaVA-NeXT (Liu et al., 2024b) have improved resolution handling and reasoning capabilities.

Commercial models have pushed the boundaries further. GPT-4V (OpenAI, 2023) demonstrates unprecedented visual understanding and reasoning, though architectural details remain proprietary. Gemini (Team et al., 2023) achieves state-of-the-art performance across numerous multimodal benchmarks through native multimodal pretraining rather than connecting separate vision and language models.

The Qwen series has emerged as a particularly strong line of multimodal models. Qwen-VL (Bai et al., 2023) introduced a versatile VLM supporting multiple languages and resolutions. Qwen2-VL (Wang et al., 2024a) significantly improved upon this with enhanced visual reasoning, video understanding, and multilingual OCR capabilities across 29 languages. The latest Qwen2.5-VL (Team, 2024) further advances the architecture with dynamic resolution support and improved instruction following, achieving state-of-the-art performance on various benchmarks while maintaining efficient inference.

Similarly, Google's Gemma family has expanded into multimodal territory. PaliGemma (Beyer et al., 2024) combines a SigLIP vision encoder with Gemma language models for versatile vision-language understanding. Gemma-2 (Team et al., 2024) improved the base architecture, leading to enhanced multimodal capabilities when combined with vision encoders. These models demonstrate strong performance while being more accessible than larger commercial offerings.

Other notable open-source alternatives include InternVL (Chen et al., 2024), which scales vision foundation models for generic visual-linguistic tasks, and the Yi-VL series (Young et al., 2024), which offers competitive performance with bilingual (Chinese-English) specialization.

These autoregressive models excel at complex reasoning, visual question answering, and generating detailed descriptions. However, they require significant computational resources during inference due to sequential token generation and cannot easily cache embeddings for retrieval tasks. Furthermore, their multilingual capabilities typically depend on the underlying LLM's language coverage, making it challenging to add new languages without extensive retraining.

The fundamental architectural differences between contrastive and autoregressive approaches lead to complementary strengths: contrastive models like CLIP excel at retrieval and classification with efficient inference, while autoregressive models provide superior reasoning and generation capabilities at higher computational cost. Our topology-aware alignment method specifically targets the unique challenges of multilingual contrastive models, where maintaining geometric consistency across languages is crucial for retrieval performance.

### A.4 Topological Analysis of the Embedding Space.

Recent studies have emphasized the importance of preserving the topological structure in representation learning (Moor et al., 2020; Trofimov et al., 2023; Zilberstein et al., 2024). Complementary efforts have employed topological representations enriching representation learning (Carrière et al., 2020; Papillon et al., 2023; Wen et al., 2024). Building on these advances, topology-aware techniques have been applied in the context of VLMs to improve embedding robustness and generalization (Zhang et al., 2024; Rahim et al., 2024; Huang, 2025). Furthermore, topological representations have proven effective for knowledge distillation and continual learning, where the latent space geometry acts as transferable knowledge (Kim et al., 2024; Wang et al., 2024b; Hai et al., 2025).

Despite these advances, topological consistency across multilingual embeddings remains underexplored. This work proposes a topological alignment framework that enforces structural coherence between the latent spaces of CLIP and MCLIP using persistent homology.

## B  PERSISTENT HOMOLOGY

Topological data analysis (TDA) characterizes the shape of data by extracting topological features that are stable to small perturbations. We assume the observed points are sampled from an unknown manifold embedded in a metric space. Given a finite point cloud $X = \{x_i\}_{i=1}^{N}$ with metric $d$, we construct a nested family of simplicial complexes (e.g., a Vietoris-Rips filtration) indexed by a scale parameter $\alpha$. Persistent homology computes homology across scales and records when features, such

as connected components and loops, are born and die. The resulting multiset of birth-death pairs is the persistence diagram. These summaries provide geometric signals.

**Point Clouds and the Vietoris-Rips Filtration.** Let $X = \{x_i\}_{i=1}^N \subset (\mathcal{X}, d)$. For $\alpha \geq 0$, the *Vietoris-Rips (Rips) complex* $\mathrm{VR}_\alpha(X)$ is the abstract simplicial complex whose $k$-simplices are all $(k+1)$-tuples $\{x_{i_0}, \ldots, x_{i_k}\}$ with pairwise distances $\max_{p,q} d(x_{i_p}, x_{i_q}) \leq \alpha$. As $\alpha$ increases, the complexes are nested

$$\mathrm{VR}_{\alpha_1}(X) \subseteq \mathrm{VR}_{\alpha_2}(X) \quad \text{for } \alpha_1 \leq \alpha_2, \tag{9}$$

yielding the Rips filtration $\{\mathrm{VR}_\alpha(X)\}_{\alpha \geq 0}$.

**Weighted Graphs and the Rips Filtration.** For a weighted graph $G = (V, E, w)$ with weight function $\omega : E \to \mathbb{R}_{\geq 0}$, we define the *Rips complex* $\mathrm{VR}_\alpha(G)$ as the abstract simplicial complex whose 1-skeleton consists of the vertex set $V$ and all edges $(u, v) \in E$ with $w(u, v) \leq \alpha$. Higher-order simplices are then included whenever all their edges are present. As $\alpha$ increases, the complexes form a nested sequence $\mathrm{VR}_{\alpha_1}(G) \subseteq \mathrm{VR}_{\alpha_2}(G)$ for $\alpha_1 \leq \alpha_2$, yielding the Rips filtration $\{\mathrm{VR}_\alpha(G)\}_{\alpha \geq 0}$ induced by the graph weights.

**Persistent Homology and Persistence Diagrams.** Fix a homological dimension $k \in \{0, 1, 2, \ldots\}$ and a coefficient field (we use $\mathbb{Z}_2$). The inclusion maps in the filtration induce homomorphism between homology groups $H_k(\mathrm{VR}_{\alpha_1}) \to H_k(\mathrm{VR}_{\alpha_2})$ for $\alpha_1 \leq \alpha_2$. Each topological feature $\eta$ (a $k$-dimensional class) *appears* (is born) at scale $b$ ($H_k(\mathrm{VR}_b)$) and *disappears* (dies) at scale $d \geq b$ ($H_k(\mathrm{VR}_d)$). The multiset of pairs $(b, d)$ is the $k$-dimensional *persistence diagram* $D_k$. For $k=0$, all components are born at $b=0$, and deaths record the merger times of components.

**Distances Between Persistence Diagrams.** Let $D_1$ and $D_2$ be persistence diagrams, and let $\Delta = \{(t, t) : t \in \mathbb{R}\}$ be the diagonal line in $\mathbb{R}^2$. We compare diagrams by allowing matches to points on $\Delta$. For $p \in [1, \infty)$, the *p-Wasserstein distance* is

$$W_p(D_1, D_2) = \left[ \inf_\gamma \sum_{u \in D_1 \cup \Delta} \left( \|u - \gamma(u)\|_p \right)^p \right]^{1/p}, \tag{10}$$

where $\gamma$ ranges over all bijections between $D_1 \cup \Delta$ and $D_2 \cup \Delta$, and $\|\cdot\|_p$ denotes $L_p$-norm. The special case $p=\infty$ yields the *bottleneck distance*

$$W_\infty(D_1, D_2) = \inf_\gamma \sup_{u \in D_1 \cup \Delta} \|u - \gamma(u)\|_\infty. \tag{11}$$

These metrics enjoy well-known stability properties: small perturbations of the input metric (or filtration function) produce small changes in the diagrams (Skraba and Turner, 2020).

**Sliced Wasserstein distance (SWD).** SWD approximates the $d$-dimensional Wasserstein distance by projecting the data onto many 1-dimensional lines and averaging the resulting one-dimensional Wasserstein costs. This yields a fast $\mathcal{O}(K N \log N)$, differentiable, and GPU-friendly objective that is well suited as a training loss. We now give the formal definition.

Given two finite point sets $X = \{x_i\}_{i=1}^N \subset \mathbb{R}^n$ and $Y = \{y_j\}_{j=1}^N \subset \mathbb{R}^n$ (uniform weights), the sliced $p$-Wasserstein distance compares them by averaging one-dimensional $p$-Wasserstein costs of their projections. For a unit direction $\theta \in S^{n-1}$, project $s_i = \langle x_i, \theta \rangle$ and $t_j = \langle y_j, \theta \rangle$, and let $s_{(1)} \leq \cdots \leq s_{(N)}$ and $t_{(1)} \leq \cdots \leq t_{(N)}$ be the sorted values. The 1D cost along $\theta$ is

$$W_p^{1D}(\theta) = \left( \frac{1}{N} \sum_{i=1}^N \left| s_{(i)} - t_{(i)} \right|^p \right)^{1/p}.$$

Averaging over directions yields

$$\mathrm{SW}_p(X, Y) = \left( \int_{S^{d-1}} \left( W_p^{1D}(\theta) \right)^p d\sigma(\theta) \right)^{1/p}$$

where $\sigma$ is the uniform measure on $S^{d-1}$. In practice, we approximate the integral with $K$ directions $\{\theta_k\}_{k=1}^K$ sampled uniformly:

$$\mathrm{SW}_p^{(K)}(X, Y) = \left( \tfrac{1}{K} \sum_{k=1}^K \left( W_p^{\mathrm{1D}}(\theta_k) \right)^p \right)^{1/p} \tag{12}$$

which can be computed in $\mathcal{O}(K\,N \log N)$ time via sorting per direction.

## C   PROOF OF THEOREM

**Theorem 1.** *Let $0 \leq \epsilon \leq 1$. Define $G_\epsilon = (V, E, \omega_\epsilon)$ by*

$$\omega_\epsilon(e) = \begin{cases} \omega(e), & \text{if } \omega(e) \leq \epsilon, \\ 1, & \text{if } \omega(e) > \epsilon. \end{cases} \tag{13}$$

*Let $m(\epsilon) := \#\{(0, d) \in D_0^{\mathrm{Rips}}(G) \mid \epsilon < d < \infty\}$, i.e., the number of finite $0$-dimensional persistence points of $G$ whose death times exceed $\epsilon$. Then*

$$W_p\big(D_0^{\mathrm{Rips}}(G),\, D_0^{\mathrm{Rips}}(G_\epsilon)\big) \ \leq \ m(\epsilon)^{1/p}\,(1 - \epsilon) \tag{14}$$

*and $0 \leq m(\epsilon) \leq N - 1$ where $W_p$ denotes $p$-Wasserstein distance.*

*Proof.* Let $\mathcal{F}_G = \{\mathrm{VR}_\alpha(G)\}_{\alpha \geq 0}$ and $\mathcal{F}_{G_\epsilon} = \{\mathrm{VR}_\alpha(G_\epsilon)\}_{\alpha \geq 0}$ denote the (graph-level) 1-skeleton filtrations where

$$\mathrm{VR}_\alpha(G) = V \cup \{\, e \in E \mid \omega(e) \leq \alpha \,\}, \qquad \mathrm{VR}_\alpha(G_\epsilon) = V \cup \{\, e \in E \mid \omega_\epsilon(e) \leq \alpha \,\}.$$

Since 0-dimensional homology is depends only on 0 and 1-simplices, it suffices to consider the filtered 1-skeleton. For $\alpha \leq \epsilon$, we have $\omega_\epsilon(e) = \omega(e)$ whenever $\omega(e) \leq \epsilon$, hence $\mathrm{VR}_\alpha(G) = \mathrm{VR}_\alpha(G_\epsilon)$. Moreover, since $\omega_\epsilon(e) \in \{\omega(e), 1\}$, for every $\alpha$ with $\epsilon < \alpha < 1$ we have $\mathrm{VR}_\alpha(G_\epsilon) = \mathrm{VR}_\epsilon(G_\epsilon)$, i.e., the filtration of $G_\epsilon$ is constant on $[\epsilon, 1)$. Consequently, in $D_0^{\mathrm{Rips}}(G_\epsilon)$ every class that is still alive at time $\epsilon$ dies precisely at $\alpha = 1$ when all remaining edges of weight 1 are added.

In 0-dimensional persistence points, all births occur at $0$, and there are $N$ points including a single essential class. Thus, points of $D_0^{\mathrm{Rips}}(G)$ with death times $d \leq \epsilon$ also appear with the same deaths in $D_0^{\mathrm{Rips}}(G_\epsilon)$, while each point with death $d \in (\epsilon, 1)$ in $D_0^{\mathrm{Rips}}(G)$ corresponds to a point with death $1$ in $D_0^{\mathrm{Rips}}(G_\epsilon)$.

Define a bijection $\gamma' : D_0^{\mathrm{Rips}}(G) \cup \Delta \to D_0^{\mathrm{Rips}}(G_\epsilon) \cup \Delta$ by

$$\gamma'(0, d) \ = \ \begin{cases} (0, d), & d \leq \epsilon, \\ (0, 1), & \epsilon < d \leq 1, \end{cases} \tag{15}$$

and map the essential class to the essential class. (No diagonal points are used here, but allowing $\Delta$ keeps the statement standard.) With the usual $\ell_p$ ground metric on $\mathbb{R}^2$, we have

$$\|(0, d) - \gamma'(0, d)\|_p \ = \ \begin{cases} 0, & d \leq \epsilon, \\ |1 - d|, & \epsilon < d \leq 1. \end{cases} \tag{16}$$

The number $m(\epsilon)$ of pairs with $\epsilon < d \leq 1$ is at most $N - 1$ (all but the essential component). Therefore,

$$\sum_{u \in D_0^{\mathrm{Rips}}(G) \cup \Delta} \left( \|u - \gamma'(u)\|_p \right)^p \ < \ m(\epsilon)\,(1 - \epsilon)^p, \tag{17}$$

and $0 \leq m(\epsilon) \leq N - 1$ since $|1 - d| < 1 - \epsilon$ for every $d \in (\epsilon, 1]$. Taking the infimum over all bijections and the $p$-th root yields

$$W_p\big(D_0^{\mathrm{Rips}}(G),\, D_0^{\mathrm{Rips}}(G_\epsilon)\big) \ < \ \big(m(\epsilon)\,(1 - \epsilon)^p\big)^{1/p} \tag{18}$$

$$= \ m(\epsilon)^{1/p}\,(1 - \epsilon), \tag{19}$$

which proves the claim. $\qquad\square$

# D DATASETS AND EXPERIMENTAL DETAILS

**Datasets.** We use the multilingual caption dataset introduced by (Carlsson et al., 2022), publicly available at `https://huggingface.co/datasets/M-CLIP/ ImageCaptions-7M-Translations`. While the corpus provides translations for multiple languages, Korean is not included. To incorporate Korean, we augment the corpus by replacing a portion of captions with Korean translations; the replacement ratio and exact sampling procedure are specified below.

Table 2: Per-language sample counts before/after adding Korean. Before: all languages except Vietnamese had 150,000; Vietnamese had 100,000; Korean was absent. Totals are preserved.

| Language | Before | After | $\Delta$ | Language | Before | After | $\Delta$ |
|---|---|---|---|---|---|---|---|
| afrikaans | 150000 | 147000 | −3000 | italian | 150000 | 147000 | −3000 |
| albanian | 150000 | 147000 | −3000 | japanese | 150000 | 147000 | −3000 |
| amharic | 150000 | 147000 | −3000 | korean | 0 | 138000 | +138000 |
| arabic | 150000 | 147000 | −3000 | macedonian | 150000 | 147000 | −3000 |
| azerbaijani | 150000 | 147000 | −3000 | malayalam | 150000 | 147000 | −3000 |
| bengali | 150000 | 147000 | −3000 | marathi | 150000 | 147000 | −3000 |
| bosnian | 150000 | 147000 | −3000 | polish | 150000 | 147000 | −3000 |
| bulgarian | 150000 | 147000 | −3000 | portuguese | 150000 | 147000 | −3000 |
| catalan | 150000 | 147000 | −3000 | romanian | 150000 | 147000 | −3000 |
| chinese_simplified | 150000 | 147000 | −3000 | russian | 150000 | 147000 | −3000 |
| chinese_traditional | 150000 | 147000 | −3000 | serbian | 150000 | 147000 | −3000 |
| czech | 150000 | 147000 | −3000 | slovenian | 150000 | 147000 | −3000 |
| danish | 150000 | 147000 | −3000 | spanish | 150000 | 147000 | −3000 |
| dutch | 150000 | 147000 | −3000 | swahili | 150000 | 147000 | −3000 |
| english | 150000 | 147000 | −3000 | swedish | 150000 | 147000 | −3000 |
| estonian | 150000 | 147000 | −3000 | tagalog | 150000 | 147000 | −3000 |
| french | 150000 | 147000 | −3000 | telugu | 150000 | 147000 | −3000 |
| german | 150000 | 147000 | −3000 | turkish | 150000 | 147000 | −3000 |
| greek | 150000 | 147000 | −3000 | turkmen | 150000 | 147000 | −3000 |
| hindi | 150000 | 147000 | −3000 | ukrainian | 150000 | 147000 | −3000 |
| hungarian | 150000 | 147000 | −3000 | uzbek | 150000 | 147000 | −3000 |
| icelandic | 150000 | 147000 | −3000 | uyghur | 150000 | 147000 | −3000 |
| indonesian | 150000 | 147000 | −3000 | vietnamese | 100000 | 100000 | 0 |
| | | | | **Total** | Before: **7000000** | After: **7000000** | $\Delta$: **0** |

**Korean Augmentation.** In the original corpus, Korean was absent; 46 languages had 150,000 captions each and Vietnamese had 100,000, totaling 7M samples. We added Korean while preserving the per-language ratios and the total size by uniformly reallocating 3,000 captions from each non-Vietnamese language to Korean. Specifically, for every language except Vietnamese (fixed at 100,000), we randomly selected 3,000 captions and replaced them with Korean translations. This results in 147,000 samples per non-Vietnamese language (down from 150,000) and 138,000 Korean samples in total ($46 \times 3{,}000$). Table 2 summarizes the per-language counts.

Korean translations were generated using the OpenAI API with a temperature setting of 0.0 to ensure deterministic and consistent translations. To handle the large-scale translation task efficiently, we implemented a batch processing pipeline with checkpoint mechanisms. The translation system processed captions in batches of 1,000 items, with automatic checkpointing every 5,000 translations to enable recovery from potential interruptions. Each translation request included explicit instructions to return only the translated text without additional formatting or explanations. Failed translation attempts were handled with exponential backoff retry logic (up to 3 attempts) to ensure robustness against transient API failures.

**Embedding Subset.** Although the full dataset contains approximately 7M samples, we rely on the 2M precomputed text embeddings released at `ImageCaptions-7M-Embeddings`. We use this

subset to train both MCLIP and ToMCLIP and verify that it is sufficient to reproduce the MCLIP performance reported in (Carlsson et al., 2022). To evaluate the model under a low-resource condition, we further subsampled 1% of the 2M samples and trained MCLIP and ToMCLIP using this reduced training set. This setup simulates scenarios where access to multilingual annotated data is severely limited.

**Models.** For multilingual text encoding, we adopt XLM-RoBERTa (Conneau et al., 2019). We use the CLIP (ViT-B/32) image encoder (Radford et al., 2021). When comparing MCLIP and ToM-CLIP, the backbone architecture, optimizer, and learning-rate schedule are identical unless otherwise noted. We set the batch size to 256, following MCLIP (Carlsson et al., 2022). ToMCLIP($L_{\text{dm}}$), ToMCLIP($L_{\text{ta}}$), and ToMCLIP denote models trained with the proposed total loss $L_{\text{total}}$ using coefficients $(\alpha, \beta, \gamma) = (1, 0.01, 0)$, $(1, 0, 0.01)$, and $(1, 0.01, 0.01)$, respectively. To construct a sparse graph, let $DM$ denote the pairwise distance matrix; we set $\epsilon = \text{mean}(DM) - 0.5 * \text{std}(DM)$, computed separately for each point cloud. For the sliced Wasserstein distance, we use $p = 2$ and average over 50 random projection directions.

**Training and Evaluation.** We train under two data regimes: full-data (all available subset entries) and a 1% low-resource setting. We report zero-shot CIFAR-100 classification across 13 languages using top-1/5/10. All preprocessing, tokenization settings, batch sizes, learning rates, and early stopping are the same as MCLIP (Carlsson et al., 2022), except for the loss function, which includes our topology-alignment objective. The results are from a single training run, consistent with standard research practices (Radford et al., 2021; Carlsson et al., 2022; Chen et al., 2023; Yang et al., 2024). For the 1% low-resource setting, this work reports the mean over three independent runs.

# E  PROMPTS OF MULTILINGUAL LANGUAGE FOR THE EVALUATION OF ZERO-SHOT CLASSIFICATION ON THE CIFAR-100

To perform zero-shot classification on the CIFAR-100 dataset, we construct language-specific text prompts to match the expected format of each language. These prompts are used to generate class-specific textual descriptions, which are then embedded using the multilingual text encoder. The general template follows the format of "a photo of a {}" in English, where the placeholder is replaced by the class name. Table 3 summarizes the prompt templates used for each language in our evaluation.

Table 3: Prompt templates used for each language in the zero-shot classification task. The placeholder {} is replaced with the class name.

| Language (ISO) | Prompt Template |
|---|---|
| English (En) | a photo of a {} |
| French (Fr) | une photo dún(e) {} |
| Spanish (Es) | una foto de un(a) {} |
| German (De) | ein Foto von einem/einer {} |
| Italian (It) | una foto di un(a) {} |
| Russian (Ru) | фото {} |
| Polish (Pl) | zdjęcie {} |
| Turkish (Tr) | {} fotoğrafı |
| Danish (Da) | et billede af en {} |
| Japanese (Ja) | {}の写真 |
| Chinese (Zh) | 一张{}的照片 |
| Korean (Ko) | {}가 있는 사진 |
| Vietnamese (Vi) | một bức ảnh về {} |

# F  ADDITIONAL RESULTS

## F.1  Connectivity and Sparsity Analysis of Approximation Method

We evaluated the effect of the threshold parameter $\epsilon$ on the sparsity and connectivity of the sparsified graph $G_\epsilon = (V, E_\epsilon)$. Across uniform and Gaussian random point clouds in $\mathbb{R}^n$ with

$N \in \{64, 128, 256, 512\}$, we measured the number of connected components $c(\epsilon)$ and the average sparsity $(|E_\epsilon| / \binom{N}{2})$ when $\epsilon = \mu - \lambda\sigma$ for $\lambda \in \{1.0, 0.5, 0, -0.5, -1.0\}$, where $\mu$ and $\sigma$ denote the mean and standard deviation of all weights in $G$.

Table 4 reveals a clear monotonic trade-off. As $\lambda$ decreases, the threshold $\epsilon = \mu - \lambda\sigma$ increases, leading to a higher edge density and fewer connected components. At a large positive value of $\lambda$, graphs are sparse but fragmented into multiple components, particularly for Gaussian point clouds, which exhibit a stronger central concentration. As $\lambda$ becomes smaller, the graphs quickly become connected ($c(\epsilon) = 1$), and sparsity rises above 0.5. At $\lambda = 0.5$, the graphs achieve near connectivity across all $N$, while retaining only about 30% of the edges. ToMCLIP adopts this setting, as it offers an effective balance between sparsity and connectivity.

Table 4: Average connected components $c(\epsilon)$ and sparsity by $\lambda$ value on random point clouds ($n = 512$, 10 trials).

| | Connected components $c(\epsilon)$ | | | | | | | | | | Sparsity | | | | | | | | | |
| | Uniform ($\lambda$) | | | | | Gaussian ($\lambda$) | | | | | Uniform ($\lambda$) | | | | | Gaussian ($\lambda$) | | | | |
| $N$ | 1.0 | 0.5 | 0.0 | $-0.5$ | $-1.0$ | 1.0 | 0.5 | 0.0 | $-0.5$ | $-1.0$ | 1.0 | 0.5 | 0.0 | $-0.5$ | $-1.0$ | 1.0 | 0.5 | 0.0 | $-0.5$ | $-1.0$ |
|---|---|---|---|---|---|---|---|---|---|---|---|---|---|---|---|---|---|---|---|---|
| 64 | 1.6 | 1.1 | 1.0 | 1.0 | 1.0 | 4.1 | 1.4 | 1.1 | 1.0 | 1.0 | 0.158 | 0.306 | 0.496 | 0.690 | 0.840 | 0.157 | 0.309 | 0.504 | 0.693 | 0.840 |
| 128 | 1.7 | 1.0 | 1.0 | 1.0 | 1.0 | 3.1 | 1.2 | 1.0 | 1.0 | 1.0 | 0.160 | 0.310 | 0.499 | 0.692 | 0.841 | 0.160 | 0.311 | 0.502 | 0.694 | 0.841 |
| 256 | 1.1 | 1.0 | 1.0 | 1.0 | 1.0 | 3.2 | 1.2 | 1.1 | 1.0 | 1.0 | 0.159 | 0.308 | 0.499 | 0.692 | 0.841 | 0.159 | 0.310 | 0.503 | 0.693 | 0.842 |
| 512 | 1.0 | 1.0 | 1.0 | 1.0 | 1.0 | 2.2 | 1.0 | 1.0 | 1.0 | 1.0 | 0.158 | 0.308 | 0.499 | 0.690 | 0.841 | 0.159 | 0.310 | 0.502 | 0.692 | 0.841 |

## F.2 Evaluation on CIFAR-100

Table 5: Average Top-$k$ accuracy (%) of the zero-shot classification on CIFAR-100 across 13 languages.

| | Low | | | Full | | |
| | Top-1 | Top-5 | Top-10 | Top-1 | Top-5 | Top-10 |
|---|---|---|---|---|---|---|
| CLIP | 20.29 | 32.47 | 39.04 | 20.29 | 32.47 | 39.04 |
| MCLIP | 30.21 | 56.67 | 67.90 | 50.72 | 76.49 | 84.93 |
| ToMCLIP($L_{dm}$) | 31.12 | 56.47 | 67.60 | 50.53 | 75.84 | 84.87 |
| ToMCLIP($L_{ta}$) | 30.45 | 57.14 | 68.26 | 50.73 | 77.12 | 85.73 |
| ToMCLIP | **31.91** | **58.15** | **69.26** | **51.32** | **77.46** | **85.81** |

Table 6: Top-1 accuracy (%) of zero-shot classification on CIFAR-100 across 13 languages (Full vs. Low).

| Setting | Model | En | Fr | Es | De | It | Ru | Pl | Tr | Da | Ja | Zh | Ko | Vi | Avg |
|---|---|---|---|---|---|---|---|---|---|---|---|---|---|---|---|
| | CLIP | **60.67** | 40.11 | 37.49 | 36.06 | 26.93 | 1.06 | 10.71 | 9.54 | 17.87 | 12.40 | 5.21 | 2.21 | 3.49 | 20.29 |
| | MCLIP | 58.86 | 49.14 | 51.13 | 51.23 | 51.13 | 49.83 | **51.40** | 51.24 | 55.13 | 33.01 | **54.70** | 51.16 | 51.35 | 50.72 |
| Full | ToMCLIP($L_{dm}$) | 57.79 | 46.19 | 50.39 | **56.13** | 50.39 | 48.62 | 50.29 | 50.99 | 56.62 | **33.85** | 52.35 | 52.28 | 51.03 | 50.53 |
| | ToMCLIP($L_{ta}$) | 58.10 | 48.67 | 48.54 | 52.42 | 51.44 | **52.67** | 50.74 | 50.57 | 57.09 | 32.86 | 51.90 | 51.37 | **53.15** | 50.73 |
| | ToMCLIP | 58.93 | **50.76** | **52.67** | 54.27 | **52.68** | 50.63 | 50.04 | 51.21 | **57.50** | 31.33 | 52.97 | **52.41** | 51.72 | **51.32** |
| | CLIP | **60.67** | 40.11 | **37.49** | **36.06** | 26.93 | 1.06 | 10.71 | 9.54 | 17.87 | 12.40 | 5.21 | 2.21 | 3.49 | 20.29 |
| | MCLIP | 35.70 | 32.40 | 29.64 | 31.20 | 28.19 | 32.21 | 27.25 | 25.05 | 33.88 | 24.41 | 33.63 | 30.38 | 28.77 | 30.21 |
| Low | ToMCLIP($L_{dm}$) | 37.84 | 33.12 | 30.32 | 31.13 | 29.82 | 32.70 | 28.87 | 25.16 | 35.24 | 25.91 | 34.32 | 31.27 | 28.82 | 31.12 |
| | ToMCLIP($L_{ta}$) | 37.79 | 31.01 | 29.75 | 31.25 | 28.82 | 32.07 | 28.18 | 24.43 | 34.49 | 23.87 | 32.79 | 30.75 | **30.67** | 30.45 |
| | ToMCLIP | 37.64 | 34.08 | 31.12 | 31.09 | **31.28** | **34.08** | 30.20 | 25.75 | 36.11 | 26.65 | 35.18 | 31.79 | 29.90 | **31.91** |

Table 7: Top-5 accuracy (%) of zero-shot classification on CIFAR-100 across 13 languages (Full vs. Low).

| Setting | Model | En | Fr | Es | De | It | Ru | Pl | Tr | Da | Ja | Zh | Ko | Vi | Avg |
|---|---|---|---|---|---|---|---|---|---|---|---|---|---|---|---|
| | CLIP | 85.26 | 58.75 | 56.94 | 55.17 | 42.02 | 6.49 | 16.71 | 17.56 | 27.47 | 25.33 | 14.26 | 6.74 | 9.44 | 32.47 |
| | MCLIP | **85.38** | 77.07 | 78.25 | 77.13 | 79.41 | 79.06 | **76.51** | 78.06 | 79.98 | 46.85 | **81.39** | 77.86 | 77.45 | 76.49 |
| Full | ToMCLIP($L_{dm}$) | 84.23 | 73.35 | 73.30 | **82.06** | 77.03 | 76.31 | 74.19 | 78.61 | 79.84 | 49.05 | 79.75 | 79.40 | 78.85 | 75.84 |
| | ToMCLIP($L_{ta}$) | 84.22 | 75.25 | 74.00 | 79.58 | 79.96 | **80.76** | 76.09 | **79.58** | 80.80 | **50.10** | 81.12 | 79.28 | **81.83** | 77.12 |
| | ToMCLIP | 84.78 | **78.87** | **79.11** | 80.97 | **80.09** | 78.39 | 74.66 | 78.89 | **81.27** | 49.58 | 80.38 | **79.79** | 80.16 | **77.46** |
| | CLIP | **85.26** | **58.75** | **56.94** | 55.17 | 42.02 | 6.49 | 16.71 | 17.56 | 27.47 | 25.33 | 14.26 | 6.74 | 9.44 | 32.47 |
| | MCLIP | 67.99 | 57.26 | 52.52 | **60.26** | 50.52 | 57.82 | 52.05 | 48.35 | 61.92 | 49.45 | 67.07 | 54.48 | 57.00 | 56.67 |
| Low | ToMCLIP($L_{dm}$) | 67.70 | 58.15 | 53.18 | 59.20 | 51.96 | 56.97 | 51.52 | 46.31 | 62.34 | 48.56 | 65.88 | 56.48 | 55.85 | 56.47 |
| | ToMCLIP($L_{ta}$) | 68.39 | 57.19 | 52.97 | 59.18 | 51.44 | 58.32 | 53.15 | 48.51 | 61.70 | 50.08 | 65.50 | 56.15 | **60.28** | 57.14 |
| | ToMCLIP | 68.75 | 58.42 | 54.09 | 60.12 | **53.73** | 59.50 | 54.07 | 49.92 | 63.29 | 51.29 | 67.36 | 56.62 | 58.74 | **58.15** |

In this section, we report Top-1 and Top-5 performance on CIFAR-100 under both the full-resource and low-resource settings, where the results for the low-resource setting are averaged over three independent runs. As shown in Tables 6 and 7, ToMCLIP outperforms MCLIP in zero-shot classification on CIFAR-100 across 13 languages. These results confirm that topology-aware alignment enhances cross-lingual consistency and robustness.

## F.3 Topological Alignment Analysis

The topological alignment objective incorporated two loss components, $L_{ta}$ and $L_{dm}$. To assess their effects on the image-text latent space, we compare CLIP, MCLIP and ToMCLIP (trained with $L_{dm}$ and $L_{ta}$). We use the same prompts for English (En) and Korean (Ko) derived from CIFAR-100 class labels as in the zero-shot evaluation (Appendix E).

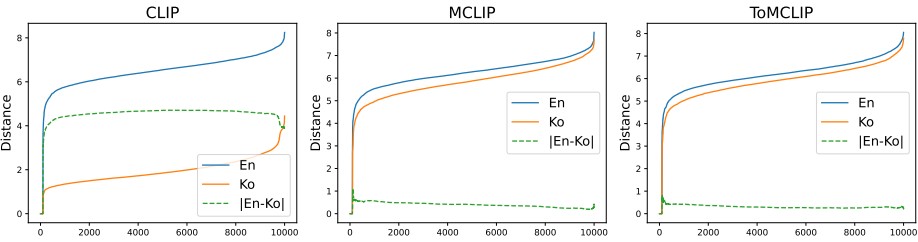

Figure 3: Sorted pairwise distance curves of English (En) vs. Korean (Ko) embeddings.

**Visualization of Shared Latent Space.**  We visualize the sorted pairwise distance curves for En and Ko embeddings. Figure 3 displays distances sorted in ascending order and the dashed green line represents the absolute pairwise distance difference $|\text{En} - \text{Ko}|$. In CLIP, a substantial discrepancy exists between the En and Ko distance distributions, reflected by a high $|\text{En} - \text{Ko}|$ curve because the CLIP model is trained using En caption datasets. The MCLIP, which is trained using multilingual data, exhibits improved alignment, significantly reducing the $|\text{En} - \text{Ko}|$ differences. Furthermore, ToMCLIP enhances the alignment, producing closer En and Ko curves. These results visually confirm that ToMCLIP achieves the highest degree of cross-lingual geometric consistency in terms of the pairwise distance, suggesting that the topological alignment loss bridges language-induced gaps.

In addition to the distance curve analysis, we provide a visualization of the shared embedding space for En and Ko CIFAR-100 class label embeddings (Figure 4). Each point represents the embedding of a prompted class label, and distances between points reflect semantic relationships in the embedding space. For each model (i.e., CLIP, MCLIP, and ToMCLIP), we project the embeddings in two dimensions using t-SNE (Maaten and Hinton, 2008) and highlight the class-level clusters. The bounding boxes were manually defined based on the En CLIP embeddings to capture coherent semantic groups, and the same grouping scheme was consistently applied to MCLIP and ToMCLIP for comparability. In CLIP, although the En embeddings form clear clusters, the Ko embeddings remain scattered, reflecting poor cross-lingual alignment. Moreover, MCLIP substantially improves alignment, with Ko embeddings aligning more closely to the manually defined clusters. Nevertheless, MCLIP still presents structural misalignment, as some clusters are mixed in the center. Furthermore, the red box overlaps with neighboring groups, and the blue box is split into two subregions, which are clearly distinguished in the En embeddings. The ToMCLIP refines this structure, producing highly consistent cross-lingual clusters that preserve the semantic grouping. The red and blue clusters become well separated from other groups in En and Ko embeddings, highlighting the robustness of the topological alignment. Conbined with the distance curve, this visualization demonstrates that ToMCLIP minimizes pairwise distance discrepancies and preserves higher-level semantic structures across languages, providing complementary evidence for the effectiveness of the proposed topological alignment loss.

**Quantitative Analysis of Shared Latent Space.**  The $L_{dm}$ term minimizes the MSE between two pairwise distance matrices. Table 8 reports the mean and RMSE of the absolute sorted pairwise distance differences ($|\text{En} - \text{Ko}|$) between En and Ko embeddings. The proposed ToMCLIP achieves substantially lower values than MCLIP, indicating improved alignment.

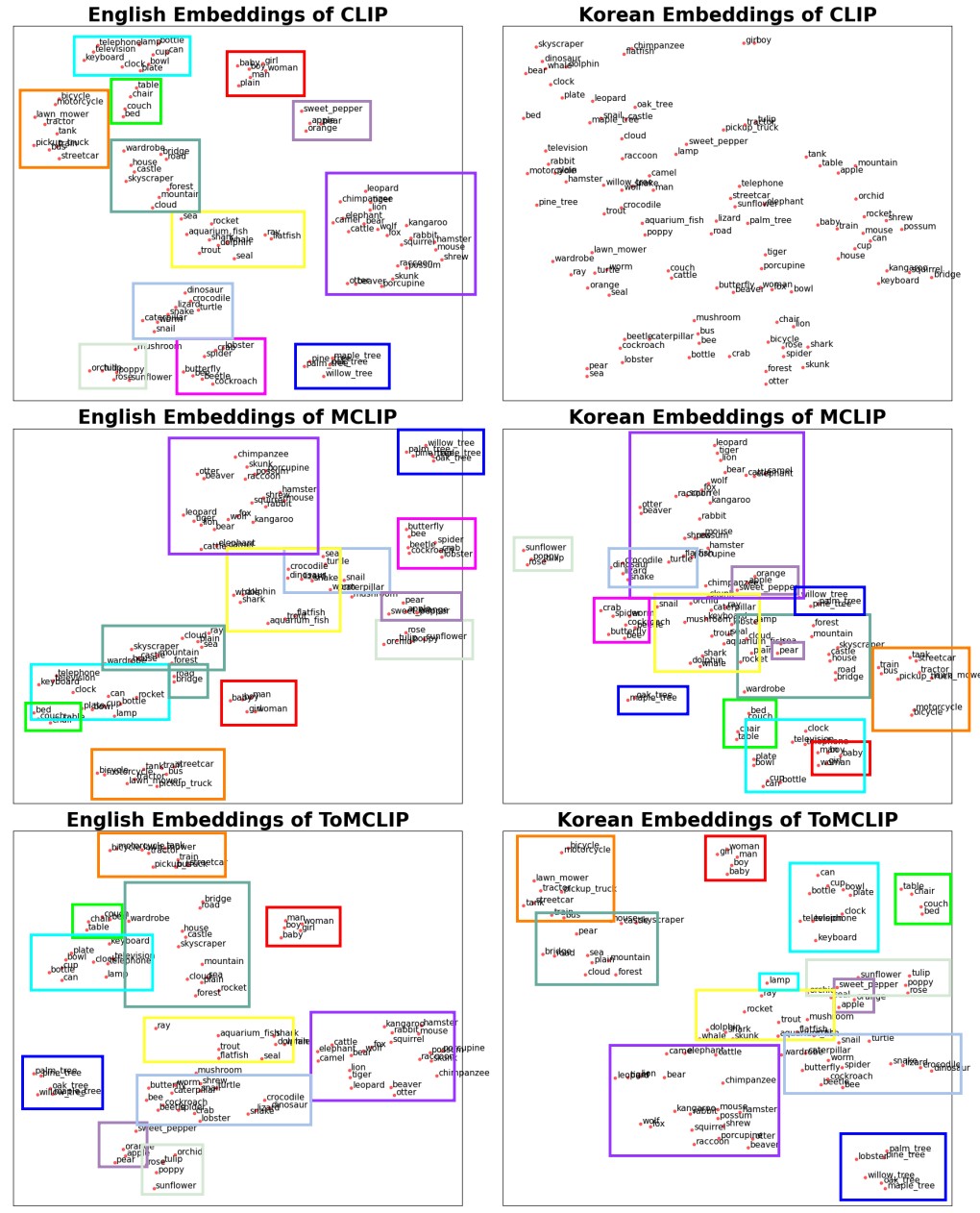

Figure 4: Two-dimensional t-SNE projections of English and Korean text embeddings

Table 8: Mean and RMSE of $|\text{En} - \text{Ko}|$.

| Model | Mean | RMSE |
|---|---|---|
| CLIP | 4.5238 | 4.5509 |
| MCLIP | 0.3920 | 0.4081 |
| ToMCLIP | **0.3050** | **0.3133** |

The $L_{\text{ta}}$ promotes topological consistency by minimizing the distance between the persistence diagrams of the two embedding sets. By the stability inequality (Eq. 4), decreasing $L_{\text{ta}}$ tightens a certified lower bound on the $p$-Wasserstein distance between the corresponding point clouds. To

verify that minimizing $L_{ta}$ yields lower $W_p^c$, Table 9 reports three metrics: $W_2^c$ (2-Wasserstein between the raw embeddings), $W_2$ (2-Wasserstein between the persistence diagrams), and $\mathrm{SW}_2^{(50)}$ (sliced 2-Wasserstein between the persistence diagrams using 50 projections). Overall, ToMCLIP yields the lowest cross-lingual distances across all metrics, confirming that topology-aware training with $L_{ta}$ enhances the topological alignment.

Table 9: Comparison of topological distances between English and Korean embeddings.

| Comparison | $W_2^c$ | $W_2$ | | $\mathrm{SW}_2^{(50)}$ | |
| --- | --- | --- | --- | --- | --- |
| | | 0-dim | 1-dim | 0-dim | 1-dim |
| CLIP (En) vs. CLIP (Ko) | 7.7870 | 34.5016 | 1.0468 | 2.8261 | 4.1593 |
| MCLIP (En) vs. MCLIP (Ko) | 2.5988 | 5.1995 | 0.9250 | 0.3670 | 0.5964 |
| ToMCLIP (En) vs. ToMCLIP (Ko) | **2.4929** | **4.2072** | **0.7444** | **0.3056** | **0.3393** |

## F.4 Ablation Study on Batch Size

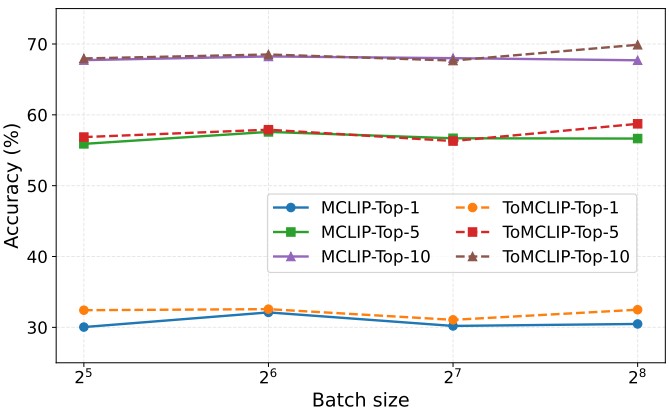

Figure 5: Ablation study on batch size in the low-resource setting.

We also investigate the effect of batch size in the low-resource setting. In our framework, the batch size corresponds to the number of sampled points considered when constructing persistence diagrams in the shared embedding space. Hence, a larger batch size allows for capturing more refined topological features and yields better approximations of the underlying geometry. However, increasing the batch size raises computational complexity, making it crucial to balance accuracy and efficiency. As shown in Figure 5, performance improves with larger batches, and we therefore adopt a batch size of 256 as the default in the experiments. We note that with batch sizes smaller than 128, the number of sampled points is insufficient to approximate the underlying data manifold, leading to limited improvements. By contrast, a batch size of 256 provides enough samples to extract topological information more effectively. Further exploration with 512 or larger batch sizes may reveal whether additional gains are possible, which we leave for future work. In addition, future work will also explore approximation techniques to further reduce computational cost while maintaining the benefits of large batch sizes.

## F.5 Ablation Study on Loss Coefficients

In Table 10, we present the ablation study on the loss coefficients under the low-resource setting. We observe that extremely large coefficients (e.g., $\beta = 0.1$ or $\gamma = 0.1$) severely degrade performance across all metrics, while small to moderate values (e.g., $\beta = 0.01, 0.001$ or $\gamma = 0.01, 0.001$) provide stable performance over the baseline. Among the tested configurations, $\beta = 0.01$ and $\gamma = 0.01$ achieve the highest scores for both Top-1 (32.49%) and Top-5 (58.73%), as well as the best Top-10 accuracy (69.89%). Therefore, we adopt $\beta = 0.01$ and $\gamma = 0.01$ as the default setting for the all experiments.

Table 10: Ablation results on loss coefficients. The experiments are conducted on **Low** resource setting.

| | Top-1 (%) | | | | Top-5 (%) | | | | Top-10 (%) | | | |
|---|---|---|---|---|---|---|---|---|---|---|---|---|
| $\beta$ \ $\gamma$ | 0.0 | 0.001 | 0.01 | 0.1 | 0.0 | 0.001 | 0.01 | 0.1 | 0.0 | 0.001 | 0.01 | 0.1 |
| 0.0 | 30.48 | 29.89 | 31.20 | 1.00 | 56.65 | 55.86 | 56.43 | 5.07 | 67.70 | 67.54 | 67.65 | 9.97 |
| 0.001 | 30.38 | 29.40 | 30.74 | 1.10 | 56.26 | 55.57 | 56.23 | 4.80 | 67.54 | 66.80 | 67.42 | 9.69 |
| 0.01 | 30.16 | 29.67 | **32.49** | 1.01 | 56.87 | 55.99 | **58.73** | 5.28 | 68.53 | 67.44 | **69.89** | 10.23 |
| 0.1 | 1.14 | 1.19 | 1.31 | 1.00 | 5.21 | 5.01 | 6.60 | 5.11 | 10.49 | 10.15 | 13.03 | 10.10 |

## F.6 Ablation on 1-Dimensional Homology

Our model uses only 0-dimensional homology ($H_0$) to extract topological features, since the birth times of 1-dimensional homology ($H_1$) features largely overlap with the pairwise distance MSE ($L_{\mathrm{dm}}$). To verify this, we empirically tested whether incorporating $H_1$ improves training. Specifically, we defined

$$L_{\mathrm{ta}} = \frac{1}{2} \mathrm{SW}_p^{(K)}\big(D_T^{(0)}, D_S^{(0)}\big) + \frac{1}{2} \mathrm{SW}_p^{(K)}\big(D_T^{(1)}, D_S^{(1)}\big), \tag{20}$$

and set $(\alpha, \beta, \gamma) = (1, 0.01, 0.01)$. We conducted experiments on **Low** setting. Adding $H_1$ lowers the overall average ($69.89 \rightarrow 69.03$), suggesting that $H_1$ provides limited additional benefit in our setting.

Table 11: Ablation on 1-dimensional homology. ToMCLIP(dim1) denotes the model trained with $L_{\mathrm{ta}}$ computed on both $H_0$ and $H_1$.

| | Languages (13) | | | | | | | | | | | | | Avg |
|---|---|---|---|---|---|---|---|---|---|---|---|---|---|---|
| Model | En | Fr | Es | De | It | Ru | Pl | Tr | Da | Ja | Zh | Ko | Vi | |
| MCLIP | 79.25 | 67.60 | 62.21 | 70.44 | 60.32 | 69.41 | 64.64 | 57.87 | 72.95 | 62.09 | 77.32 | 64.72 | 71.24 | 67.70 |
| ToMCLIP | 81.06 | **70.66** | **64.25** | 72.70 | **63.54** | **71.88** | **67.04** | **60.87** | **74.77** | **64.21** | **78.33** | 67.23 | 71.99 | **69.89** |
| ToMCLIP(dim1) | **82.27** | 69.73 | 64.02 | **73.72** | 61.16 | 71.87 | 64.53 | 60.11 | 73.58 | 60.08 | 78.07 | 66.44 | 71.81 | 69.03 |

## F.7 Effect of the Approximation Threshold for the Persistence Diagram

We control graph sparsity with a distance threshold $\epsilon = \mu - \lambda\sigma$, where $\mu$ and $\sigma$ denote the mean and standard deviation of pairwise distances, respectively; we keep edges with distance $\leq \epsilon$. We conducted an ablation study on $\lambda \in \{1.5, 1, 0.5, 0\}$ in the low-resource setting. As $\lambda$ increases, $\epsilon$ decreases and the graph becomes sparser, which reduces memory/time but may remove informative structure. Table 12 summarizes the results. Increasing $\lambda$ makes the graph sparser and speeds up persistence diagram computation ($0.075\,\mathrm{s} \rightarrow 0.011\,\mathrm{s}$ from $\lambda=0$ to $1.5$), but excessive sparsity hurts accuracy. As the graph becomes denser (smaller $\lambda$), the persistence diagram approximation approaches the exact persistence diagram and accuracy does not decrease. In practice, $\lambda = 0.5$ already makes the approximation error negligible. Choosing $\lambda < 0.5$ increases computation without yielding further gains, whereas $\lambda > 0.5$ introduces additional sparsity, incurs approximation error, and lowers accuracy. Consistent with the analysis in Section F.1, the persistence diagram approximation error near $\lambda = 0.5$ is negligible, which supports adopting $\lambda = 0.5$ as the default balance between performance and computational cost.

| $\lambda$ | Time(s) | En | Fr | Es | De | It | Ru | Pl | Tr | Da | Ja | Zh | Ko | Vi | Avg |
|---|---|---|---|---|---|---|---|---|---|---|---|---|---|---|---|
| 1.5 | **0.01119** | 78.73 | 66.95 | 62.78 | 69.67 | 60.10 | 68.71 | 62.82 | 57.38 | 70.80 | 57.57 | 75.08 | 65.76 | 71.67 | 66.77 |
| 1 | 0.02434 | 80.50 | 68.89 | 63.22 | 71.75 | 61.96 | 69.80 | 63.59 | 60.82 | 73.06 | 59.97 | 77.87 | 68.12 | 71.82 | 68.57 |
| 0.5 | 0.04608 | **81.06** | **70.66** | **64.25** | **72.70** | **63.54** | **71.88** | **67.04** | **60.87** | **74.77** | **64.21** | **78.33** | 67.23 | **71.99** | **69.89** |
| 0 | 0.07519 | 80.42 | 70.13 | 63.32 | 70.21 | 61.91 | 71.67 | 65.51 | 59.19 | 73.78 | 61.91 | 78.21 | **68.20** | 71.84 | 68.95 |

Table 12: Top-10 accuracy (%) of zero-shot classification on CIFAR-100 across 13 languages.

## F.8 Effect of the Number of Projections for SWD

We approximate the SWD in Eq. 12 via Monte Carlo sampling with $K$ random projection directions. The computational complexity is $\mathcal{O}(K\,N\log N)$; since runtime grows approximately linearly with $K$, we ablate $K$ to select a balanced value. Table 13 reports the ablation in the low-resource setting.

Empirically, increasing $K$ improves accuracy up to a point: the average Top-10 accuracy rises from 66.99 ($K$=5) and 66.81 ($K$=10) to 68.16 ($K$=30), peaking at 69.89 ($K$=50). For $K$=50 and $K$=100, performance is similar while the computational cost roughly doubles; hence we adopt $K$=50 as the default.

| $K$ | En | Fr | Es | De | It | Ru | Pl | Tr | Da | Ja | Zh | Ko | Vi | Avg |
|---|---|---|---|---|---|---|---|---|---|---|---|---|---|---|
| 5 | 78.04 | 66.92 | 60.42 | 69.17 | 59.95 | 71.64 | 63.43 | 55.38 | 72.69 | 58.69 | 77.32 | 65.25 | 71.93 | 66.99 |
| 10 | 78.03 | 66.08 | 61.88 | 70.07 | 60.06 | 69.05 | 63.45 | 57.91 | 71.41 | 58.00 | 76.50 | 64.61 | 71.51 | 66.81 |
| 30 | 79.47 | 67.18 | 63.51 | 70.57 | 60.48 | 69.53 | 66.41 | 58.76 | 72.10 | 59.19 | 77.84 | **67.92** | 73.13 | 68.16 |
| 50 | **81.06** | 70.66 | 64.25 | **72.70** | 63.54 | **71.88** | **67.04** | **60.87** | **74.77** | **64.21** | **78.33** | 67.23 | 71.99 | **69.89** |
| 100 | 79.58 | **71.81** | **66.18** | 71.97 | **64.61** | 70.92 | 64.30 | 57.44 | 73.07 | 58.33 | 78.17 | 66.63 | **73.30** | 68.95 |

Table 13: Top-10 accuracy (%) of zero-shot classification on CIFAR-100 across 13 languages.

### F.8.1 Multilingual Image-Text Retrieval on xFlickr&CO

This work evaluates multilingual image-text retrieval on xFlickr&CO (Bugliarello et al., 2022) across eight languages (En, Es, De, Id, Ru, Tr, Ja, Zh). The benchmark comprises 2K images (1K from Flickr30K and 1K from MSCOCO), each paired with a single parallel caption in all eight languages, enabling evaluation of both retrieval directions. This work presents the results for image retrieval (**IR**; text→image) and text retrieval (**TR**; image→text). Following standard practice, we compute recall at $K$ (R@K, $K \in \{1, 5, 10\}$) and average the scores across languages. For each language, R@K is evaluated over 2,000 queries.

Table 14 summarizes the language-averaged results and Table 15 breaks down R@1 by language. In both tables, red ▲ ( blue ▼) marks improvement (degradation) over MCLIP under the same settings and direction. In the **Full** regime, ToMCLIP($L_{dm}$), ToMCLIP($L_{ta}$), and ToMCLIP yield consistent average gains over MCLIP for **IR** and **TR** across all metrics (R@1,5, and 10). In the more challenging **Low** regime, they also achieve consistent average gains over MCLIP. These results indicate that the proposed losses improve cross-lingual alignment in the shared embedding space.

### F.9 Training Time and Evaluation Time

To assess computational efficiency, we compared the average training time per epoch between the two models, MCLIP and ToMCLIP. We trained with one NVIDIA A100 (80 GB) on a single-node server (2× AMD EPYC 7513). The baseline MCLIP required approximately 285 minutes per epoch, whereas the proposed ToMCLIP, which incorporates the additional topology loss and distance matrix alignment, required 357 minutes per epoch. Although ToMCLIP increases the training cost relative to MCLIP, the additional overhead remains manageable considering the substantial improvement in cross-lingual alignment performance. This is made possible by our persistence diagram approximation strategy, which employs MST-based computation and graph sparsification to avoid the exponential complexity of constructing full Rips complexes.

Importantly, evaluation time remains unchanged between MCLIP and ToMCLIP. Since our method only modifies the training objective and does not alter the model architecture, no additional computation is introduced during inference. Thus, both models share identical evaluation speed and memory requirements, ensuring that the performance gains of ToMCLIP come at no cost during deployment.

## G  ADDITIONAL RESULTS WITH VIT-B/16 PLUS CLIP IMAGE ENCODER

We replace the CLIP image encoder with ViT-B/16+ (Cherti et al., 2023), which is trained on the LAION-400M dataset (Schuhmann et al., 2021). The multilingual text encoder remains XLM-RoBERTa (Conneau et al., 2019), as in our main experiments. Except for the image backbone, the entire training and evaluation setup is identical to the setup described earlier.

For data, we use the publicly released precomputed text embeddings from `ImageCaptions-7M-Embeddings`, which contains 7M caption embeddings compatible with the ViT-B/16+ (by contrast, the corresponding ViT-B/16 release provides about 2M embeddings). All ViT-B/16+ runs use the full 7M set; under the low-resource condition, we uniformly subsample 1% of these ($\sim$70K samples).

Table 14: Multilingual retrieval on xFlickr&CO. Average R@k (%) across 8 languages (Low vs. Full). ▲ indicates an improvement over MCLIP (same setting and direction), ▼ indicates a decrease.

| Direction | Model | Low | | | Full | | |
|---|---|---|---|---|---|---|---|
| | | R@1 | R@5 | R@10 | R@1 | R@5 | R@10 |
| IR | CLIP | 12.08 | 22.12 | 27.19 | 12.08 | 22.12 | 27.19 |
| | MCLIP | 33.51 | 62.04 | 73.70 | 50.13 | 77.51 | 85.86 |
| | ToMCLIP($L_{dm}$) | 34.49 (▲0.98) | 62.93 (▲0.89) | **74.50** (▲0.80) | **50.85** (▲0.72) | **78.25** (▲0.74) | **86.56** (▲0.70) |
| | ToMCLIP($L_{ta}$) | **34.50** (▲0.99) | **62.96** (▲0.93) | 74.45 (▲0.74) | 50.79 (▲0.66) | 78.01 (▲0.50) | 86.19 (▲0.33) |
| | ToMCLIP | 34.03 (▲0.52) | 62.59 (▲0.56) | 74.00 (▲0.30) | 50.76 (▲0.63) | 77.99 (▲0.48) | 86.48 (▲0.62) |
| TR | CLIP | 16.01 | 28.75 | 35.40 | 16.01 | 28.75 | 35.40 |
| | MCLIP | 39.39 | 68.02 | 78.65 | 53.38 | 79.48 | 87.34 |
| | ToMCLIP($L_{dm}$) | 39.71 (▲0.32) | 68.63 (▲0.61) | 79.38 (▲0.74) | 54.01 (▲0.63) | **80.38** (▲0.90) | **88.08** (▲0.74) |
| | ToMCLIP($L_{ta}$) | **40.29** (▲0.90) | **69.18** (▲1.16) | **79.61** (▲0.97) | 53.83 (▲0.45) | 79.91 (▲0.43) | 87.80 (▲0.46) |
| | ToMCLIP | 39.51 (▲0.12) | 68.42 (▲0.40) | 78.96 (▲0.32) | **54.07** (▲0.69) | 79.98 (▲0.50) | 87.67 (▲0.33) |

Table 15: Multilingual retrieval on xFlickr&CO. R@1 retrieval accuracy (%) across languages. ▲ and ▼ mark improvements/decreases over MCLIP for the same setting and direction; here only the icons are shown.

| Setting | Direction | Model | Languages | | | | | | | | Avg |
|---|---|---|---|---|---|---|---|---|---|---|---|
| | | | En | Es | De | Id | Ru | Tr | Ja | Zh | |
| Full | IR | CLIP | 54.90 | 22.05 | 11.00 | 4.15 | 0.35 | 1.90 | 1.95 | 0.35 | 12.08 |
| | | MCLIP | 55.00 | 54.65 | 48.45 | 48.95 | 56.65 | 53.35 | 35.45 | 48.50 | 50.12 |
| | | ToMCLIP($L_{dm}$) | 55.10 ▲ | 55.10 ▲ | 48.65 ▲ | 49.50 ▲ | 56.95 ▲ | 54.35 ▲ | 38.20 ▲ | 48.95 ▲ | 50.85 ▲ |
| | | ToMCLIP($L_{ta}$) | 55.40 ▲ | 54.95 ▲ | 49.15 ▲ | 49.15 ▲ | 57.35 ▲ | 53.50 ▲ | 38.20 ▲ | 48.65 ▲ | 50.79 ▲ |
| | | ToMCLIP | 55.60 ▲ | 55.15 ▲ | 48.40 ▼ | 50.00 ▲ | 56.70 ▲ | 53.70 ▲ | 38.00 ▲ | 48.55 ▲ | 50.76 ▲ |
| | TR | CLIP | 58.55 | 29.10 | 17.15 | 10.80 | 0.80 | 4.25 | 5.25 | 2.15 | 16.01 |
| | | MCLIP | 58.60 | 58.90 | 48.95 | 51.45 | 61.15 | 55.05 | 39.55 | 53.35 | 53.38 |
| | | ToMCLIP($L_{dm}$) | 59.20 ▲ | 59.35 ▲ | 49.25 ▲ | 51.80 ▲ | 61.05 ▼ | 56.50 ▲ | 40.75 ▲ | 54.15 ▲ | 54.01 ▲ |
| | | ToMCLIP($L_{ta}$) | 58.50 ▼ | 60.15 ▲ | 49.70 ▲ | 51.70 ▲ | 60.90 ▼ | 55.20 ▲ | 40.70 ▲ | 53.80 ▲ | 53.83 ▲ |
| | | ToMCLIP | 59.55 ▲ | 59.25 ▲ | 49.55 ▲ | 53.70 ▲ | 61.55 ▲ | 54.85 ▼ | 40.70 ▲ | 53.40 ▼ | 54.07 ▲ |
| Low | IR | CLIP | 54.90 | 22.05 | 11.00 | 4.15 | 0.35 | 1.90 | 1.95 | 0.35 | 12.08 |
| | | MCLIP | 37.05 | 35.72 | 30.08 | 36.00 | 38.30 | 30.17 | 27.87 | 32.88 | 33.51 |
| | | ToMCLIP($L_{dm}$) | 37.85 ▲ | 37.27 ▲ | 30.65 ▲ | 37.40 ▲ | 39.98 ▲ | 31.05 ▲ | 28.17 ▲ | 33.53 ▲ | 34.49 ▲ |
| | | ToMCLIP($L_{ta}$) | 38.00 ▲ | 36.65 ▲ | 31.23 ▲ | 36.55 ▲ | 39.60 ▲ | 31.27 ▲ | 29.17 ▲ | 33.50 ▲ | 34.50 ▲ |
| | | ToMCLIP | 37.10 ▲ | 37.23 ▲ | 30.55 ▲ | 36.37 ▲ | 38.85 ▲ | 30.15 ▼ | 28.48 ▲ | 33.52 ▲ | 34.03 ▲ |
| | TR | CLIP | 58.55 | 29.10 | 17.15 | 10.80 | 0.80 | 4.25 | 5.25 | 2.15 | 16.01 |
| | | MCLIP | 42.15 | 42.83 | 35.17 | 41.85 | 44.38 | 36.57 | 33.10 | 39.07 | 39.39 |
| | | ToMCLIP($L_{dm}$) | 42.55 ▲ | 42.48 ▼ | 35.93 ▲ | 42.33 ▲ | 45.72 ▲ | 36.32 ▼ | 32.92 ▼ | 39.47 ▲ | 39.71 ▲ |
| | | ToMCLIP($L_{ta}$) | 43.77 ▲ | 43.37 ▲ | 35.90 ▲ | 43.13 ▲ | 46.03 ▲ | 36.70 ▲ | 33.27 ▲ | 40.17 ▲ | 40.29 ▲ |
| | | ToMCLIP | 42.92 ▲ | 43.07 ▲ | 35.02 ▼ | 41.98 ▲ | 45.17 ▲ | 36.20 ▼ | 32.65 ▼ | 39.05 ▼ | 39.51 ▲ |

**CIFAR-100 Zero-Shot Classification.** Replacing the image backbone with ViT-B/16+ preserves the main trend: topology-aware objectives improve multilingual zero-shot accuracy over MCLIP in both regimes (Table 16 and 17). On the **Full** setting, ToMCLIP($L_{ta}$) attains the best averages (Top-1/5/10 = 66.18/86.35/90.89) improving over MCLIP (64.54/85.30/89.99) by **+1.64/+1.05/+0.90** points, respectively. On the **Low** setting, the combined ToMCLIP model yields the highest averages (53.31/74.88/82.01) surpassing MCLIP (50.24/73.50/81.17) by **+3.07/+1.38/+0.84**. Notably, $L_{ta}$ alone also improves alignment quality under Low (51.42/74.47/81.97). These results are consistent with the main paper: enforcing topological consistency via $L_{ta}$ strengthens cross-lingual alignment in the shared embedding space.

Table 16: Average Top-$k$ accuracy (%) of the zero-shot classification on CIFAR-100 across 13 languages.

| | Low | | | Full | | |
|---|---|---|---|---|---|---|
| | Top-1 | Top-5 | Top-10 | Top-1 | Top-5 | Top-10 |
| CLIP | 24.39 | 35.91 | 42.47 | 24.39 | 35.91 | 42.47 |
| MCLIP | 50.24 | 73.50 | 81.17 | 64.54 | 85.30 | 89.99 |
| ToMCLIP($L_{dm}$) | 52.33 | 74.68 | 81.84 | 65.92 | 85.88 | 90.44 |
| ToMCLIP($L_{ta}$) | 51.42 | 74.47 | 81.97 | **66.18** | **86.35** | **90.89** |
| ToMCLIP | **53.31** | **74.88** | **82.01** | 65.53 | 85.82 | 90.33 |

**Multilingual Image–Text Retrieval on xFlickr&CO.** With the ViT-B/16+ image encoder, topology-aware objectives improve multilingual retrieval over MCLIP in most settings (Table 18). On **Full**, ToMCLIP($L_{ta}$) attains the best averages for both directions (**IR**: R@1/5/10 = **62.98/85.79/91.60** vs. MCLIP: 62.24/85.27/91.09 and **TR**: **63.79/86.21/91.98** vs. 62.82/85.47/91.32). On **Low**, the

Table 17: Top-$k$ accuracy (%) of zero-shot classification on CIFAR-100 across 13 languages (Full vs. Low). ViT-B/16+ is used for CLIP image encoder.

| Setting | Model | En | Fr | Es | De | It | Ru | Pl | Tr | Da | Ja | Zh | Ko | Vi | Avg |
|---|---|---|---|---|---|---|---|---|---|---|---|---|---|---|---|
| | | | | | | | | Top-1 accuracy (%) | | | | | | | |
| | CLIP | 72.81 | 52.12 | 45.49 | 46.15 | 40.49 | 4.43 | 11.57 | 12.37 | 19.96 | 3.68 | 2.41 | 1.19 | 4.46 | 24.39 |
| | MCLIP | 72.42 | 66.85 | 69.25 | 56.04 | 69.85 | 67.57 | 64.14 | 65.87 | 69.86 | 38.09 | 69.11 | 66.28 | 63.75 | 64.54 |
| Full | ToMCLIP($L_{dm}$) | **73.24** | **67.76** | 68.90 | 64.60 | 69.30 | **68.16** | **66.39** | **69.54** | 70.28 | 38.00 | **69.48** | 66.98 | 64.34 | 65.92 |
| | ToMCLIP($L_{ta}$) | 72.21 | 67.31 | **69.61** | **69.30** | 68.50 | 67.30 | 64.74 | 67.93 | **70.39** | **41.10** | 68.75 | **67.51** | **65.69** | **66.18** |
| | ToMCLIP | 72.92 | **67.76** | 69.31 | 67.04 | **70.76** | 67.25 | 64.05 | 68.80 | 69.72 | 37.91 | 68.01 | 63.64 | 64.77 | 65.53 |
| | CLIP | **72.81** | 52.12 | 45.49 | 46.15 | 40.49 | 4.43 | 11.57 | 12.37 | 19.96 | 3.68 | 2.41 | 1.19 | 4.46 | 24.39 |
| | MCLIP | 58.56 | 52.47 | 52.93 | 53.94 | 47.82 | 53.18 | 48.25 | 45.04 | 50.76 | 39.18 | 53.17 | 49.73 | 48.13 | 50.24 |
| Low | ToMCLIP($L_{dm}$) | 62.66 | 54.16 | 54.11 | 54.45 | 49.45 | 55.88 | 49.43 | 47.59 | 52.62 | 41.04 | 56.23 | 52.41 | **50.32** | 52.33 |
| | ToMCLIP($L_{ta}$) | 62.46 | 54.78 | 53.41 | 54.75 | 49.09 | 50.94 | 49.22 | 45.68 | 52.68 | 39.26 | 56.01 | 51.16 | 48.97 | 51.42 |
| | ToMCLIP | 63.58 | **55.88** | **54.59** | **57.61** | **49.96** | **56.66** | **50.31** | **49.26** | **54.08** | **41.40** | **56.41** | **53.22** | 50.13 | **53.31** |
| | | | | | | | | Top-5 accuracy (%) | | | | | | | |
| | CLIP | 92.84 | 72.59 | 62.72 | 64.43 | 56.85 | 11.12 | 19.36 | 21.52 | 28.42 | 10.89 | 7.99 | 7.19 | 10.85 | 35.91 |
| | MCLIP | 92.81 | 87.94 | 90.49 | 82.12 | **89.68** | 88.43 | 83.43 | 88.71 | 87.86 | 50.84 | **90.96** | 88.57 | 87.04 | 85.30 |
| Full | ToMCLIP($L_{dm}$) | 93.20 | **88.48** | **90.50** | 84.67 | 89.26 | 87.72 | **85.38** | 89.81 | **88.01** | 52.14 | 90.24 | 88.77 | 88.26 | 85.88 |
| | ToMCLIP($L_{ta}$) | 93.04 | 88.01 | 89.32 | **89.76** | 89.30 | 87.78 | 83.99 | 89.84 | 87.99 | **55.97** | 90.37 | **89.02** | 88.21 | **86.35** |
| | ToMCLIP | **93.65** | 88.44 | 90.45 | 87.66 | 89.64 | **88.70** | 83.47 | **89.95** | 87.96 | 49.97 | **90.96** | 86.73 | 88.12 | 85.82 |
| | CLIP | **92.84** | 72.59 | 62.72 | 64.43 | 56.85 | 11.12 | 19.36 | 21.52 | 28.42 | 10.89 | 7.99 | 7.19 | 10.85 | 35.91 |
| | MCLIP | 83.44 | 73.41 | 73.25 | 78.28 | 66.15 | 77.10 | 70.11 | 67.66 | 72.80 | 60.21 | 81.99 | 75.26 | 75.79 | 73.50 |
| Low | ToMCLIP($L_{dm}$) | 85.33 | 74.40 | 73.40 | 77.95 | 66.20 | **78.64** | **71.62** | 72.08 | 74.47 | 59.85 | 82.26 | **77.26** | 77.38 | 74.68 |
| | ToMCLIP($L_{ta}$) | 84.97 | **74.63** | 74.56 | 79.33 | 66.21 | 76.99 | 71.23 | 69.11 | **75.00** | 60.28 | 82.80 | 75.45 | **77.52** | 74.47 |
| | ToMCLIP | 85.11 | 73.72 | **74.71** | **80.35** | **66.46** | 77.66 | 70.76 | **72.17** | 74.98 | **60.57** | **83.40** | 76.62 | 76.97 | **74.88** |
| | | | | | | | | Top-10 accuracy (%) | | | | | | | |
| | CLIP | 96.32 | 79.39 | 71.42 | 72.38 | 64.10 | 18.04 | 25.92 | 27.40 | 34.99 | 18.04 | 13.90 | 13.53 | 16.71 | 42.47 |
| | MCLIP | 96.41 | 92.03 | 94.25 | 89.52 | 93.35 | 92.51 | 88.74 | 93.26 | 92.10 | 56.07 | **95.35** | **94.28** | 91.96 | 89.99 |
| Full | ToMCLIP($L_{dm}$) | 96.65 | 92.69 | **94.68** | 89.92 | 93.35 | 92.72 | **90.17** | 94.00 | 91.31 | 58.17 | 94.65 | 94.26 | 93.15 | 90.44 |
| | ToMCLIP($L_{ta}$) | 96.53 | 92.01 | 93.76 | **93.61** | **93.58** | 92.36 | 88.83 | **94.44** | **92.23** | **62.28** | 94.90 | 94.17 | 92.89 | **90.89** |
| | ToMCLIP | **96.72** | **92.90** | 94.14 | 92.45 | 93.40 | **93.70** | 88.01 | 94.14 | 91.54 | 55.21 | 95.15 | 93.43 | **93.47** | 90.33 |
| | CLIP | **96.32** | 79.39 | 71.42 | 72.38 | 64.10 | 18.04 | 25.92 | 27.40 | 34.99 | 18.04 | 13.90 | 13.53 | 16.71 | 42.47 |
| | MCLIP | 91.16 | 79.26 | 81.08 | 85.98 | 72.05 | 84.87 | 79.89 | 77.03 | 80.56 | 67.11 | 89.01 | 82.35 | 84.89 | 81.17 |
| Low | ToMCLIP($L_{dm}$) | 91.16 | 80.00 | 80.47 | 86.07 | 72.30 | **86.25** | **80.06** | 81.06 | 82.00 | 66.52 | 88.75 | **83.52** | **85.75** | 81.84 |
| | ToMCLIP($L_{ta}$) | 91.65 | **80.80** | **83.27** | 86.57 | **73.19** | 85.57 | 78.75 | 77.75 | 81.99 | **68.89** | 89.70 | 82.31 | 85.13 | 81.97 |
| | ToMCLIP | 91.54 | 79.59 | 81.88 | **87.54** | 72.80 | 85.15 | 79.71 | **80.38** | **82.13** | 67.88 | **89.79** | 83.41 | 84.37 | **82.01** |

combined ToMCLIP variant yields the top averages for IR (R@1/5/10 = **58.53/83.37/90.51**), while ToMCLIP($L_{dm}$) is strongest for TR (R@1/5/10 = **57.99/83.84/90.63**). These trends mirror our zero-shot CIFAR-100 results: enforcing topological consistency via $L_{ta}$ improves cross-lingual alignment.

Table 18: Multilingual retrieval on xFlickr&CO. Average R@k (%) across 8 languages (Low vs. Full). ▲ indicates an improvement over MCLIP (same setting and direction), ▼ indicates a decrease.

| Direction | Model | Low | | | Full | | |
|---|---|---|---|---|---|---|---|
| | | R@1 | R@5 | R@10 | R@1 | R@5 | R@10 |
| IR | CLIP | 16.38 | 27.00 | 32.06 | 16.38 | 27.00 | 32.06 |
| | MCLIP | 56.44 | 82.28 | 89.60 | 62.24 | 85.27 | 91.09 |
| | ToMCLIP($L_{dm}$) | 57.91 (▲1.47) | 83.15 (▲0.87) | 90.37 (▲0.77) | 62.24 (▲0.00) | 85.39 (▲0.12) | 91.22 (▲0.13) |
| | ToMCLIP($L_{ta}$) | 57.58 (▲1.14) | 82.77 (▲0.49) | 90.12 (▲0.53) | **62.98** (▲0.74) | **85.79** (▲0.52) | **91.60** (▲0.51) |
| | ToMCLIP | **58.53** (▲2.08) | **83.37** (▲1.09) | **90.51** (▲0.91) | 61.91 (▼0.33) | 84.89 (▼0.38) | 90.78 (▼0.31) |
| TR | CLIP | 18.91 | 31.46 | 36.59 | 18.91 | 31.46 | 36.59 |
| | MCLIP | 56.73 | 83.33 | 90.34 | 62.82 | 85.47 | 91.32 |
| | ToMCLIP($L_{dm}$) | **57.99** (▲1.26) | **83.84** (▲0.51) | **90.63** (▲0.29) | 62.95 (▲0.13) | 85.67 (▲0.20) | 91.14 (▼0.17) |
| | ToMCLIP($L_{ta}$) | 57.33 (▲0.60) | 83.26 (▼0.06) | 90.27 (▼0.07) | **63.79** (▲0.97) | **86.21** (▲0.74) | **91.98** (▲0.66) |
| | ToMCLIP | 57.57 (▲0.84) | 83.39 (▲0.06) | 90.61 (▲0.28) | 62.19 (▼0.63) | 85.09 (▼0.38) | 90.84 (▼0.47) |

Table 19: Multilingual retrieval on xFlickr&CO. R@1 retrieval accuracy (%) across languages. ▲ and ▼ mark improvements/decreases over MCLIP for the same setting and direction; here only the icons are shown.

| Setting | Direction | Model | Languages | | | | | | | | Avg |
|---|---|---|---|---|---|---|---|---|---|---|---|
| | | | En | Es | De | Id | Ru | Tr | Ja | Zh | |
| **Full** | **IR** | CLIP | 64.70 | 34.70 | 21.35 | 5.65 | 0.90 | 2.70 | 0.40 | 0.65 | 16.38 |
| | | MCLIP | 65.50 | **69.05** | 59.60 | 61.40 | **72.45** | **66.90** | 41.40 | 61.60 | 62.24 |
| | | ToMCLIP($L_{dm}$) | 65.60▲ | 68.70▼ | 59.65▲ | 61.70▲ | 72.30▼ | 65.70▼ | 42.35▲ | 61.90▲ | 62.24▲ |
| | | ToMCLIP($L_{ta}$) | 65.20▼ | 69.00▼ | **60.00**▲ | **63.05**▲ | 72.35▼ | 65.75▼ | **46.25**▲ | 62.25▲ | **62.98**▲ |
| | | ToMCLIP | **65.75**▲ | 68.60▼ | 59.20▲ | 61.30▼ | 72.20▼ | 66.00▼ | 39.90▼ | **62.35**▲ | 61.91▼ |
| | **TR** | CLIP | 66.70 | 40.45 | 26.05 | 10.05 | 1.15 | 5.10 | 0.85 | 0.90 | 18.91 |
| | | MCLIP | 68.30 | 68.90 | 59.20 | 62.00 | 73.55 | 66.75 | 42.50 | 61.35 | 62.82 |
| | | ToMCLIP($L_{dm}$) | 68.30▲ | **70.00**▲ | 57.70▼ | 62.20▲ | **73.75**▲ | 66.85▲ | 43.50▲ | 61.30▼ | 62.95▲ |
| | | ToMCLIP($L_{ta}$) | 68.80▲ | 69.85▲ | **59.75**▲ | **62.40**▲ | **73.75**▲ | 66.60▼ | **46.70**▲ | 62.45▲ | **63.79**▲ |
| | | ToMCLIP | **68.85**▲ | 69.30▲ | 57.85▼ | 61.30▼ | 72.65▼ | **67.30**▲ | 39.55▼ | 60.75▼ | 62.19▼ |
| **Low** | **IR** | CLIP | **64.70** | 34.70 | 21.35 | 5.65 | 0.90 | 2.70 | 0.40 | 0.65 | 16.38 |
| | | MCLIP | 59.05 | 60.30 | 52.45 | 55.85 | 63.45 | 55.15 | 49.30 | 56.00 | 56.44 |
| | | ToMCLIP($L_{dm}$) | 59.50▲ | **63.05**▲ | **55.30**▲ | **57.05**▲ | 64.80▲ | 55.80▲ | **49.75**▲ | 58.05▲ | 57.91▲ |
| | | ToMCLIP($L_{ta}$) | 59.50▲ | 61.95▲ | 53.75▲ | 56.80▲ | 65.80▲ | 55.80▲ | 49.65▲ | 57.40▲ | 57.58▲ |
| | | ToMCLIP | 60.55▲ | 62.80▲ | 55.25▲ | 57.00▲ | **66.60**▲ | **57.40**▲ | 49.55▲ | **59.05**▲ | **58.53**▲ |
| | **TR** | CLIP | **66.70** | 40.45 | 26.05 | 10.05 | 1.15 | 5.10 | 0.85 | 0.90 | 18.91 |
| | | MCLIP | 60.35 | 61.05 | 51.85 | 56.40 | 63.55 | 54.70 | 49.05 | 56.90 | 56.73 |
| | | ToMCLIP($L_{dm}$) | 61.45▲ | 61.70▲ | **53.45**▲ | **57.20**▲ | **65.45**▲ | **55.85**▲ | **51.30**▲ | **57.55**▲ | **57.99**▲ |
| | | ToMCLIP($L_{ta}$) | 61.45▲ | 61.35▲ | 53.10▲ | 56.25▼ | 64.40▲ | 55.70▲ | 50.00▲ | 56.40▼ | 57.33▲ |
| | | ToMCLIP | 61.15▲ | **61.80**▲ | 52.80▲ | 57.10▲ | 65.30▲ | 55.45▲ | 49.70▲ | 57.30▲ | 57.57▲ |

