# OpenReview forum: "Topological Alignment of Shared Vision-Language Embedding Space"
_NeurIPS.cc/2025/Workshop/UniReps — UniReps2025_

### Official Review · Reviewer_Ts7i · 2025-09-05
**Aligning vision-language embeddings using topological notions**

**Confidence:** 4

**Review:**

Previous works that study the alignment of multilingual CLIP models do not take into account the topology, thus leading to visibly different embedding spaces. The work proposes a topological aligment loss that explicitly penalizes the deviations of topological properties as evaluated in a minibatch, relying on an approximation of the persistence diagram. Empirical results are provided, demonstrating that the topological variant often yield superior accuracies above the baselines on both CIFAR-100 zero-shot classification and multilingual retrieval on xFlickr&CO.

Pro:

1. The work is generally well-motivated,
2. There are reasonable empirical validations for a workshop submission.

Cons:

1. Figure 4 is hard to analyze visually,
2. While allowing for reduced computational complexity, restricting the persistence diagrams to only 0-dimensional features and the birth times of the 1-dimensional features naturally lead to quite some loss of the topological structure. Especially for future work, it would be beneficial to analyze its implications.

Based on the above, I think the paper is a good fit to the venue and recommend acceptance.

**Score:**

4

**Topic Fit:**

2

---

### Official Review · Reviewer_sM2P · 2025-09-08
**Review for Topological Alignment of Shared Vision-Language Embedding Space**

**Confidence:** 5

**Review:**

Summary
     This paper aims to address the limitation of CLIP based method due to the biased toward English only data. This work proposed three different losses to align the embedding of CLIP text encoder and that of multilingual CLIP text encoder. The chosen dataset for evaluation is questionable and some section in the paper requires more clarification.

Strength: The paper has strong motivation and has interesting application

Weakness.
1.	The dataset chosen in this paper is Cifar100. It is unclear why we need VLM to perform classification for Cifar100 dataset. On the other way, Cifar100 might not be the right dataset for evaluate the proposed method.
2.	What is point-wise alignment? There is no definition of “point-wise alignment” in the main paper.
3.	Not sure why there is a section about “Approximating the Persistence Diagram” and how it can be used in the proposed method.
4. In addition, when comparing Table 1 and Table2, the performance of the proposed method (ToMCLIP) degrade after performing training under a low resource setting and its perofmrance is even worse than that of zero-shot classification. The author is suggested to provide more clarification about why this is happening.

**Score:**

2

**Topic Fit:**

2

---

### Official Review · Reviewer_ntnY · 2025-09-12
**Review of ToMCLIP: preserving structures in multilingual representations.**

**Confidence:** 4

**Review:**

**Summary**

The paper proposes ToMCLIP, a topology-aware alignment framework for multilingual CLIP. Beyond pointwise distillation from English CLIP, it adds a topology loss (sliced-Wasserstein distance on persistence diagrams) and a distance-matrix loss to preserve global and local geometry of language embedding spaces. An MST-based approximation makes persistence tractable. Experiments show modest but consistent gains in multilingual CIFAR-100 classification and xFlickr&CO retrieval in low-resource (1% of the data) training.

**Strength**
1. Clear motivation: instance-level distillation ignores structure.
2. Novel use of persistence diagrams as a structural regularizer; efficient MST-based approximation.
3. Empirical improvements with under 1% data.
4. Clearly written and relevant to the scope of the workshop.

**Weakness**

1. No comparison to simpler geometry-preserving baselines (Procrustes, MMD, Laplacian).
2. The evaluation is limited. used mostly prompt-based CIFAR-100 and small retrieval benchmarks, no free-form captions.
3. The performance gains are modest at full data scale, while strong under 1% data. Analyses of the effect plateaus with training size is lacking.
4. Anchoring all languages to English CLIP might introduce bias.
5. The approximation only took care of 0 and 1 dimensions, while sensitivity analyses are missing for higher-order features.
6. Biological plausibility is metaphorical, not mechanistic. It's hard to implement in the neuroscience context.
7. No consideration of manifold-aware alignment, which could preserve nonlinear structure more faithfully. The embeddings are treated as Euclidean point clouds; If multilingual embeddings live on nonlinear manifolds, then enforcing Euclidean distance preservation (or MST topology) could be misleading.

**Ratings**

- Originality: 4/5
- Clarity: 4/5
- Technical soundness: 3/5
- Quality of results: 3/5
- Significance: 2/5

**Score:**

3

**Topic Fit:**

3

---

### Official Review · Reviewer_73mm · 2025-09-15
**A nice application of TDA concepts to multilingual VLMs**

**Confidence:** 4

**Review:**

## Summary

Vision-Language Models (VLMs) are an indispensable component of modern computer vision, yet mainstay models are predominantly biased towards the English language due to the scarcity of multilingual multimodal datasets. The paper proposes ToMCLIP (Topological Alignment for Multilingual CLIP), a topology-aware training framework that aligns the shard vision-language embedding space using metrics on the persistent homology of the latent space. Alignment is achieved through a sparse sliced Wasserstein metric on the latent spaces. The authors approximate the embedding homology via sparse graphs to maintain applicability to the scale of modern vision language models.

## Strengths

- The authors clearly present their thesis with supporting evidence: alignment for multilingual multi-modal vision language models frequently fails. Analysis for the Korean language is emphasized.
- While persistent homology-based regularization has been proposed for e.g., pre-training of VLMs [1], the application applied to multi-lingual encoders appears novel.
- Quantitative and qualitative evidence suggest the method indeed improves alignment and functions as intended. Nevertheless, the performance improvements are modest (in comparison to the non-PH baseline MCLIP).
- Overall, this is a nice idea worth presenting as an extended abstract.

## Weaknesses

- The sliced Wasserstein distance is a fairly standard approach to comparing persistence diagrams. Constructing the persistence diagram on a sparse graph is nice for computational tractability, but otherwise, the technical novelties of the paper are rather limited.
- The approach seems to exhibit some catastrophic forgetting for the English language, as evident by the low-resource setting tables 6-8. Notably, this is not as evident for the “normal” Top-5 accuracy.
- The authors extensively validate the multi-lingual alignment with the Korean language, but this language is not included in the multi-lingual datasets. The authors include it themselves, by translating with the OpenAI api. While this is certainly a worthwhile endeavor, it sparks concern about possibly confounding the dataset generation and analysis for a specific language. The paper would significantly benefit from a more thorough analysis of the improved alignment impact of ToMCLIP for some of the other languages besides Korean.

1: Zhang, H., Zhang, L., Zhang, Y., & Mao, Z. (2024). Homology
consistency constrained efficient tuning for vision-language models. *Advances in Neural Information Processing Systems*, *37*, 93011-93032.

**Score:**

3

**Topic Fit:**

2